# Variability of the lunar semidiurnal tidal amplitudes in the ionosphere over Brazil

Ana Roberta Paulino[1], Fabiano da Silva Araújo[1], Igo Paulino[2], Cristiano Max Wrasse[3], Lourivaldo Mota Lima[1], Paulo Prado Batista[3], and Inez Staciarini Batista[3]

[1]Departamento de Física, Universidade Estadual da Paraíba, Campina Grande, Brazil
[2]Unidade Acadêmica de Física, Universidade Federal de Campina Grande, Campina Grande, Brazil
[3]Divisão de Aeronomia, Instituto Nacional de Pesquisas Espaciais, São José dos Campos, Brazil

**Correspondence:** Ana Roberta Paulino (arspaulino@gmail.com)

**Abstract.** The variability in the amplitudes of the lunar semidiurnal tide was investigated using maps of Total Electron Content over Brazil from January 2011 to December 2014. Long period variability showed that the annual variation is always present in all investigated magnetic latitudes and it represents the main component of the temporal variability. Semiannual and terannual (∼120 days) oscillations were the second and third components, respectively, but they presented significant temporal and spatial variability without a well-defined pattern. Among the short period oscillations in the amplitude of the lunar tide, the most pronounced ones were concentrated between 7-11 days. These oscillations were stronger around the equinoxes, in particular between September and November in almost all latitudes. In some years, as in 2013 and 2014, for instance, they appeared with large power spectral density in the winter hemisphere. These observed short period oscillations could be a result of a direct modulation of the lunar semidiurnal tide by planetary waves from the lower atmosphere or/and due to electrodynamics coupling of E and F region of the ionosphere.

## 1 Introduction

Planetary waves are produced by large-scale perturbations in the atmosphere that can have horizontal wavelength up to 40,000 km around the equator. Those waves can have periods which vary from a couple of days to weeks. Planetary waves are responsible for most of the temporal and spatial variation in the stratosphere and they also contribute substantially to the variability of the mesosphere and lower thermosphere (MLT). Basically, planetary waves have been classified in three types: (1) quasistationary midlatitude Rossby waves, (2) normal modes and (3) equatorial waves (Smith and Perlwitz, 2015).

Quasistationary Rossby waves are important to the midlatitude dynamics because they can largely influence the atmospheric fields like wind and temperature and they are responsible for the distribution of the ozone and other trace gases. Rossby normal modes, also known as free modes are predicted by the theory as oscillatory solutions of the Laplace's tidal equation without forcing. The Laplace's theory is constructed over an isothermal and non-damping atmosphere, thus, the real conditions of the atmosphere can produce normal modes with some similarities to the theoretical ones. The class of planetary waves which occur near the equator and the most commonly observed in the MLT region are the Kelvin waves, which are classified as low Kelvin

waves (periods of 10-15 days), fast Kelvin waves (periods of 6-10 days) and ultrafast Kelvin waves (periods of 2.5-6 days) (Chen and Miyahara, 2012).

Dissipative processes act significantly in the upward propagation of planetary waves in the atmosphere producing a pronounced damping above 100 km altitude. Among several mechanisms, the cooling by emission of heat and interaction with small-scale waves have been pointed out as the most important (e.g., Smith and Perlwitz, 2015, and references therein). However, in the last decades, large number of studies have shown evidences of oscillations with periods compatible with planetary-waves in the thermosphere-ionopshere (e.g., Forbes, 1996; Pancheva and Laštovička, 1998; Pancheva et al., 2002; Laštovička,

2006; Abdu et al., 2006, 2015; Jonah et al., 2015; Gan et al., 2015; Mo and Zhang, 2020, and references therein). Understanding how planetary waves can penetrate into the thermosphere-ionosphere system have raised up as one of the current most important topics of research in the atmospheric layer's coupling. Recent works have given some insights in this topic (e.g., Forbes et al., 2014; Gasperini et al., 2017), but further observations and investigations are necessary in order to understanding this coupling.

The lunar semidiurnal tide with period of $\sim$12.424 solar hours is the most important Moon's oscillation for the atmosphere in terms of amplitudes. Although the generation of the lunar semidiurnal tides comes from the lower levels of the atmosphere due to the Moon's gravitational attraction and interaction with vertical motion of the oceans and solid Earth, it can propagate into the thermosphere with less influence of the dissipative process. As the source of the lunar tide is well known, variation associated to the sources are predictable. Then, it can be used as an important trace to observe changes in the atmosphere as

it propagates vertically. Furthermore, modulation of the lunar semidiurnal tidal amplitudes by planetary waves can be used to explain the presence of these waves in the thermosphere-ionosphere system, which is the main purpose of the present work. Additionally, variability of long period in the lunar semidiurnal tidal amplitudes is also investigated.

Data from a network of Global Navigation Satellite System (GNSS) receivers over Brazil was used to calculate the amplitudes and phases of the lunar semidiurnal tide in the Total Electron Content (TEC) of the ionosphere from 2011 to 2014

(Paulino et al., 2017). In the present work, the temporal variability of the lunar semidiurnal tidal amplitudes was extensively investigated showing long ($>$ 60 days) and short ($<$ 25 days) period oscillations.

## 2  Analysis and results

### 2.1  Determination of the lunar tide in TEC maps

The determination of the lunar semidiurnal tides in TEC maps was done according to the Pedatella and Forbes (2010) method-

ology and only quiet days were considered ($K_p < 3$) in the analysis. After eliminate the geomagnetic influences, a Fourier analysis was performed to extract the subharmonics of the solar day (diurnal, semidiurnal and terdiurnal oscillations). Effects of the solar rotation was removed using a 27-day window moving it forward one day at time to calculate the mean solar day centered in the window. In addition, relative residual was determined dividing the residual variation of TEC by the TEC average.

In the relative residual data, a least square analysis in a window of 29-day was applied using the following equation:

$$y(\tau) = \sum_{n=1}^{3} A_n \cos\left(n\tau + \phi_n\right) \tag{1}$$

where $\tau$ if the lunar time given by $\tau = t - \nu$, $\nu$ in the age of the Moon, which is set to be $0$ at the New Moon. The solar time is represented by $t$, the amplitudes and phases of the lunar tide components are represented by $A_n$ and $\phi_n$, respectively.

## 2.2 Filtering

Further description of the methodology to calculate the TEC maps over Brazil was provided by Takahashi et al. (2016). TEC maps have also been used to calculate the amplitude and phases of the lunar semidiurnal tide from 2011 to 2014 over Brazil (Paulino et al., 2017). In the present study, the variability due to low and high frequencies in the lunar semidiurnal tide amplitudes observed in those TEC maps is investigated in details.

Figure 1 shows the filtering process in the amplitudes of the lunar semidiurnal tide calculated at $10^o$ S (magnetic).

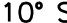

**10° S**

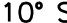

**Figure 1.** (a) Amplitudes of the lunar semidiurnal tide from 2011 to 2014 calculated at $10^o$ S (magnetic latitude). (b) Low frequencies calculated using a Butterworth kernel filter from amplitudes of panel (a). (c) Same as Figure (b), but for high frequencies.

Figure 1(a) shows the raw amplitudes in TEC units from 2011 to 2014. One can see that there are low and high frequency oscillations in the amplitudes retrieved from the TEC. Figure 1(b) shows the filtered amplitudes considering periods greater

than 30 days and Figure 1(c) shows the high frequencies greater than 1/30 days$^{-1}$. This filtering process was done using Butterworth Kernel low pass filter of order 1. Mathematically this filter can be written as:

$$filter = \frac{1}{\sqrt{1 - \left(\frac{\Omega}{\Omega_c}\right)^{2n}}} \tag{2}$$

where $\Omega$ is the frequency, $\Omega_c$ is the cutoff frequency and $n$ is the order (Roberts and Roberts, 1978). This filter is applied to the signal in the domain of the frequency and then, the filtered signal is recovered to the domain of the time.

Another important point to be analysed is the possible influence of the semidiurnal solar tide in the present results since this oscillation is very close to the semidiurnal lunar tide. TEC data collected from February to April 2014 at 30$^o$S and 54$^o$W were used to validade the present analysis. Figure 2 shows the original TEC (Panel a), amplitudes of the semidiurnal solar tide (Panel

b) , Lomb-Scargle periodogram for the amplitudes of the solar tide (Panel c), amplitudes of the semidiurnal lunar tide (Panel d) and Lomb-Scargle periodogram for the amplitudes of the lunar tide. This time interval was chosen because it present a strong quasi 8-day oscillation in the amplitudes of the semidiurnal lunar tide, which is going to be discussed in ahead. Furthermore, this is geographic coordinates corresponds to one of the most southern points of Brasil, where, in general, the solar semidiurnal tide is strong compared to the equatorial latitudes.

Figure 2 (a) shows a roughly oscillation around 8-day in the whole data, besides the well pronounced diurnal (24 h) and semidiurnal (12 h) oscillations. Figure 2 (b) shows that there is no significative oscillation of quasi 8 days in the amplitude of the solar semidiurnal tide and it is confirmed in Figure 2 (c). Additionally, Figure 2 (d) shows that the amplitude of the semidiurnal lunar tide has a well-defined quase 8-day oscillation, in which appeared above the confidence level in Figure 2 (e). These results are very relevant because they show that there is no leakage from the semidiurnal solar tide in the present results

and the proposed methodology satisfactorily separates the solar and lunar components.

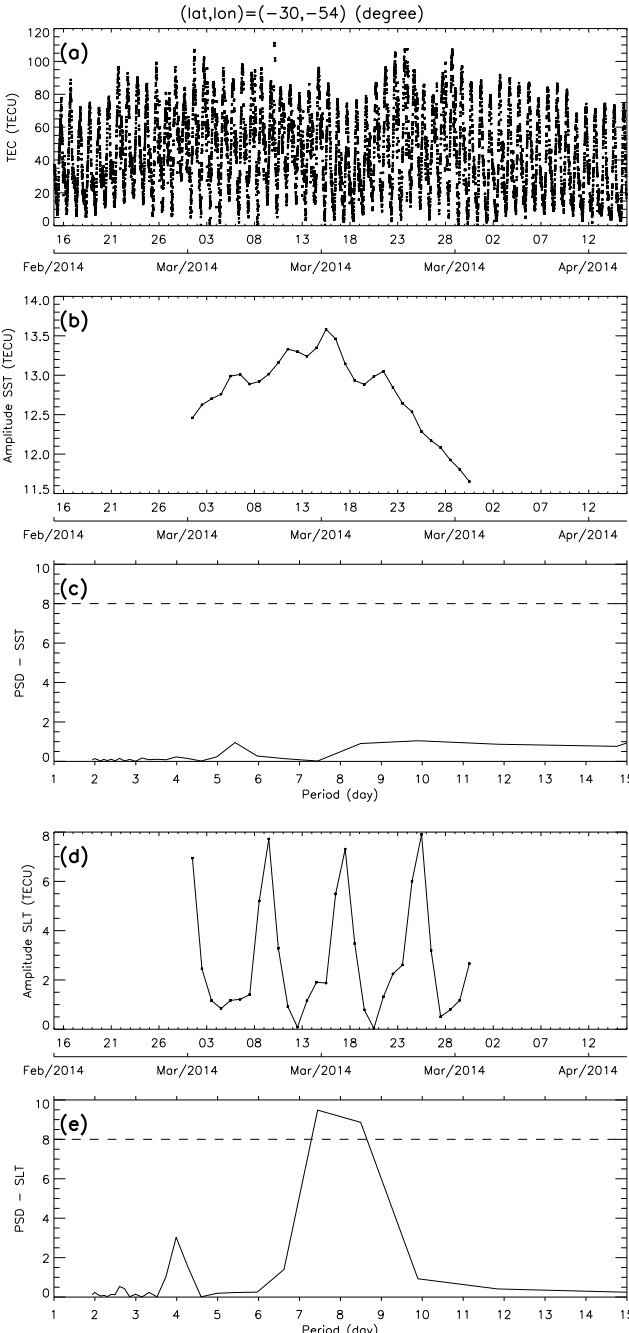

**Figure 2.** (a) TEC collected at 30°S and 54°W from February to April 2014. (b) Amplitudes of the semidiurnal solar tide calculated from the panel (a). (c) Periodogram for the amplitudes of the semidiurnal solar tide. (d) Amplitudes of the semidiurnal lunar tide calculated from the panel (a). (e) Periodogram for the amplitudes of the SLT. Horizontal dashed line represent the confidence level of 99%.

## 2.3 Seasonal and terannual variability

Considering the filtered amplitudes for periods longer than 30 days, a spectral analysis was done and the results are shown in Figure 3. The Lomb-Scargle periodograms (Lomb, 1976; Scargle, 1982) were calculated using the filtered amplitudes from 2011 to 2014 for the magnetic latitudes $10^o$ N, $0^o$, $10^o$ S and $20^o$ S.

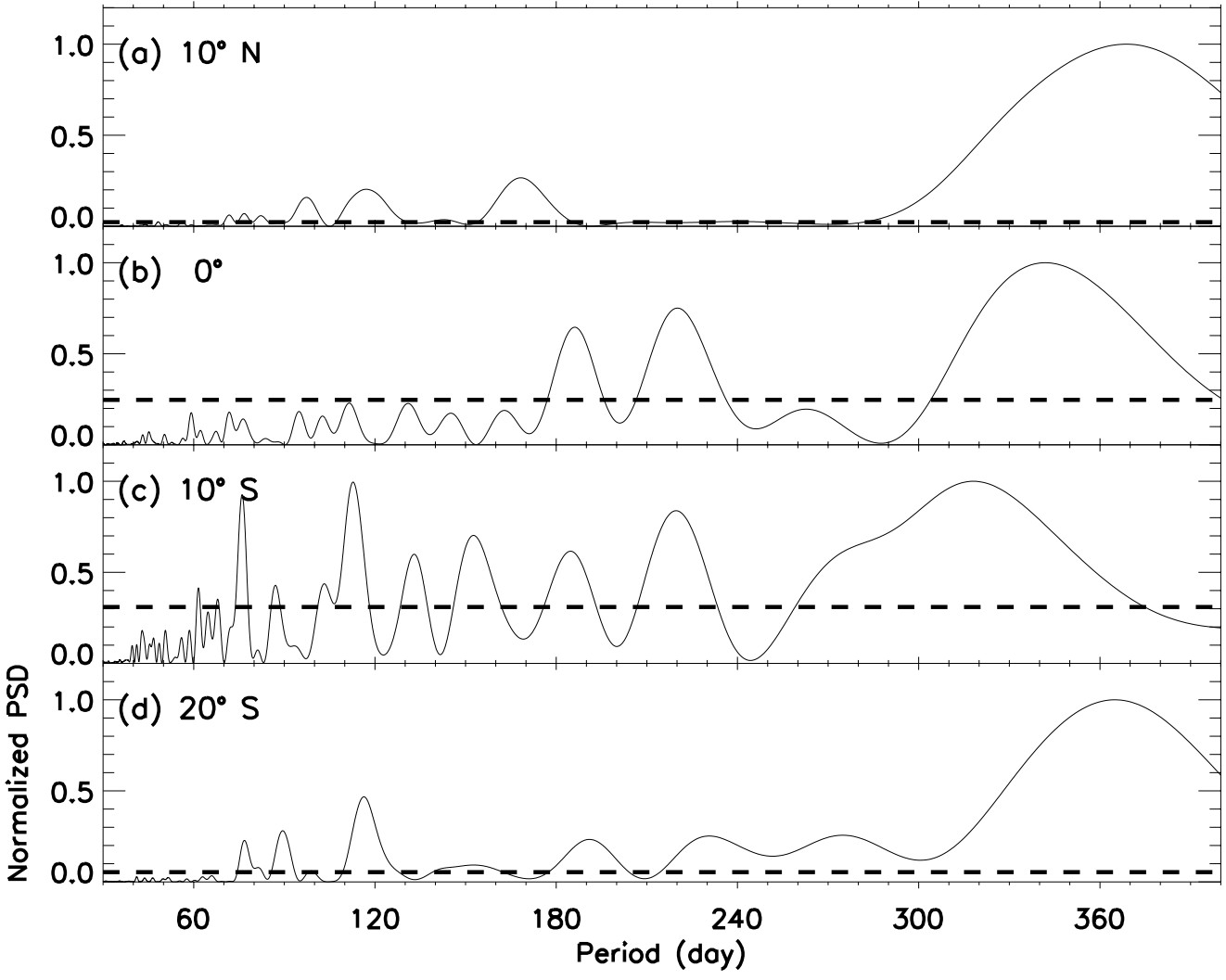

**Figure 3.** Lomb-Scargle periodogram for (a) $10^o$ N, (b) $0^o$, (c) $10^o$ S and (d) $20^o$ S. The horizontal dashed lines represent the confidence level of 99%, i.e., false alarm probability of 0.01.

Figure 3(a) shows strong the power density spectrum (PSD) associated to the annual and semiannual variations. One can also observe that there is a third peak around 120 days, i.e., terannual variation. Similar patterns to what is observed at $10^o$ N can be observed at $10^o$ S (Figure 3(c)) and $20^o$ S (Figure 3(d)) as well. At $0^o$ (Figure 3(b)), annual and semiannual variations

were strong, additionally the terannual variation was weak compared to the other latitudes. Comparing $10^o$ N to $10^o$ S, it is clear that there are more significant peak of oscillation in the South, indicating that the long oscillations are not symmetrical with respect to the magnetic equator.

In order to investigate when the periodicities shown in Figure 3 appear more frequently in the dataset, a wavelet analysis was performed and the results are shown in Figure 4 with the respective magnetic latitudes. These wavelet charts were calculated based on the methodology of Torrence and Compo (1998).

Figure 4 shows that the annual variation is always present in the amplitude of the lunar tide. Figure 4(a) shows that the semiannual variation was present in the two first years and the terannual variation appear more pronounced in the beginning of 2013, which can be composed by oscillations from 80 to 120 days. In the beginning of 2014, the terannual variation appeared as well.

Figure 4(b) (magnetic equator) shows that the semiannual variation were strong in the end of 2012 and beginning of 2013 with outspread of this peak to over 200 days. Figure 3(b) shows also this behavior in the Lomb-Scargle chart. One can also observe short oscillations of 70-80 days in the beginning of 2013 and 2014.

Figure 4(c) shows that the semiannual oscillation in the amplitude of the lunar tide becomes stronger than the annual in the second half of 2013. It is important to observe that the terannual oscillation was present in the amplitude of the lunar tide from 2011 to March 2013 and became very strong at the end of this time range, compared to the other latitudes and times. Figure 4 (c) shows also oscillation with periods shorter than 100 days along the whole observed time.

Figure 4(d) shows that the terannual oscillation appeared in 2013 and 2014 and, at this magnetic latitude, the semiannual oscillation was weaker than the terannual. Oscillations of 70-80 days have the same occurrence observed at $10^o$ N.

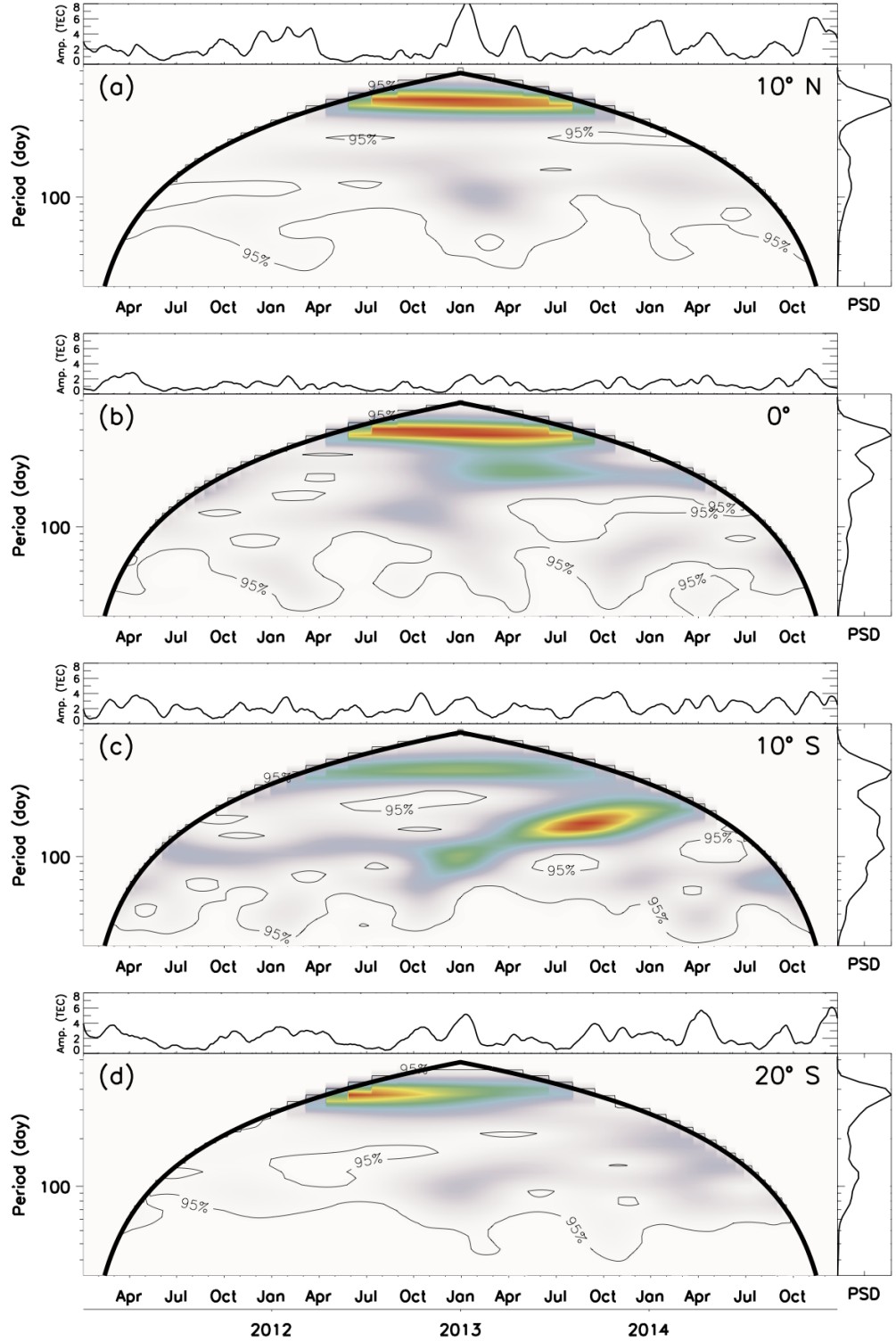

**Figure 4.** Wavelet analysis for (a) $10^o$ N, (b) $0^o$, (c) $10^o$ S and (d) $20^o$ S. The heavy black lines in the contours represent the cone of influence. The light black lines show confidence levels of 95%.

## 2.4 Short period oscillations

The short period oscillations observed in the amplitudes of the lunar tide were also investigated in this work. These oscillation are important because they can be associated with planetary waves revealing relevant aspects in the atmosphere-ionosphere coupling from below.

Figure 5 shows the Lomb-Scargle periodogram for the same latitudes used in Figure 3, but considering only periods shorter than 25 days. These periodograms were calculated using the high frequencies in the amplitudes of the lunar semidiurnal tide as exemplified in Figure 1(c). Figure 5(a) to 5(d), which represent the magnetic latitudes from $10^o$ N to $20^o$ S, show significant periodicities of 7-12 days from 2011 to 2014. Hereafter, we are going to refer to these oscillations as quasi 8 days (Q8D) oscillation, although sometimes they can be either shorter or longer than 8 days. Similar assumption has been used by Ahlquist (1982). Please, note that other long periods were also observed above the significance levels, but they were more sporadic than the Q8D oscillation as it will be shown ahead.

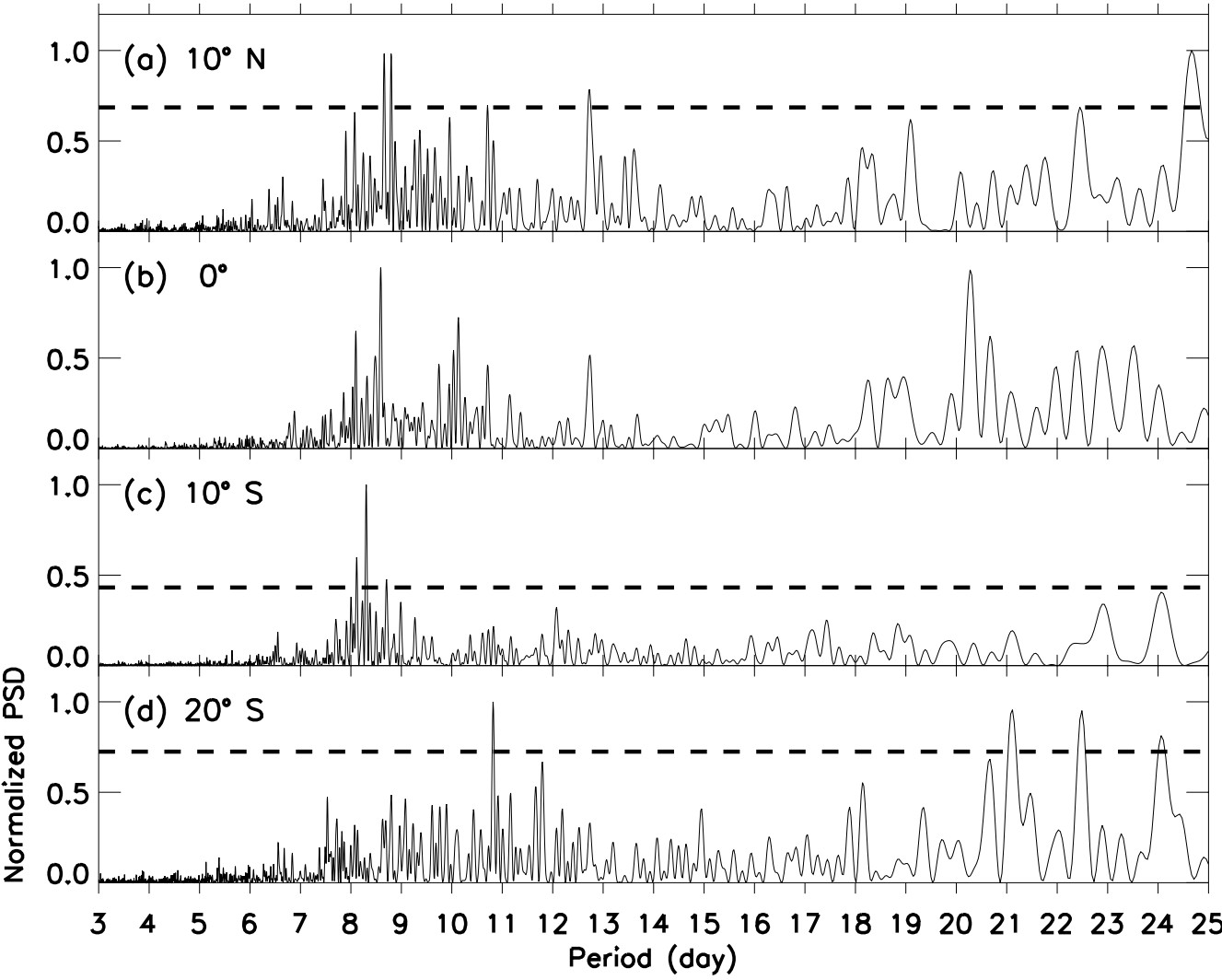

**Figure 5.** Same as Figure 2, but considering only periods shorter than 25 days. Note that at $0^o$ all periodicities were below of the confidence level since the amplitudes were smaller compared to the other latitudes.

In order to further invertigate the temporal evolution of the short period oscillation in the amplitudes of the lunar semidiurnal tide, wavelet analysis was performed for each year of observation and the results are shown in Figures 6, 7, 8 and 9. One can observe that the dominant oscillation is the Q8D along the whole period of observation. Some particularities are also observed in each year, mainly regarding the epoch of the year in which the Q8D wave is stronger.

Figure 6 shows the wavelet results for the amplitudes of the lunar semidiurnal tide in 2011. The Q8D was stronger from September to November in almost all latitudes. There was a secondary peak of this oscillation from the middle February up to April, except at $10^o$ S.

Figure 7 shows the results for 2012. Again the Q8D oscillation was the most important oscillation, but it appeared more frequently along the year, in special, out of the magnetic equator. One different aspect was the Q8D strong in February at $10^o$ N and in May at $10^o$ S.

In 2013 (Figure 8), the strength of the Q8D oscillation was more concentrated in few months. At $10^o$ N, the Q8D oscillation had more power spectral density in January and February. At $0^o$ and $10^o$ S, the oscillation appeared with more intensity from later October to December. At $20^o$ S there were two peaks of the Q8D oscillation in April and May.

Figure 9 shows the power spectral density contour for 2014. One can observe a regular behavior of the Q8D oscillation with two peaks around the equinox months.

An important result revealed from the observations is that the Q8D oscillation is always present in the equinox months. From September to November in almost all latitudes and years it was the dominant oscillation. One can also observe, that the Q8D oscillation appear strong during the winter in 2012 and 2013 for some magnetic latitudes.

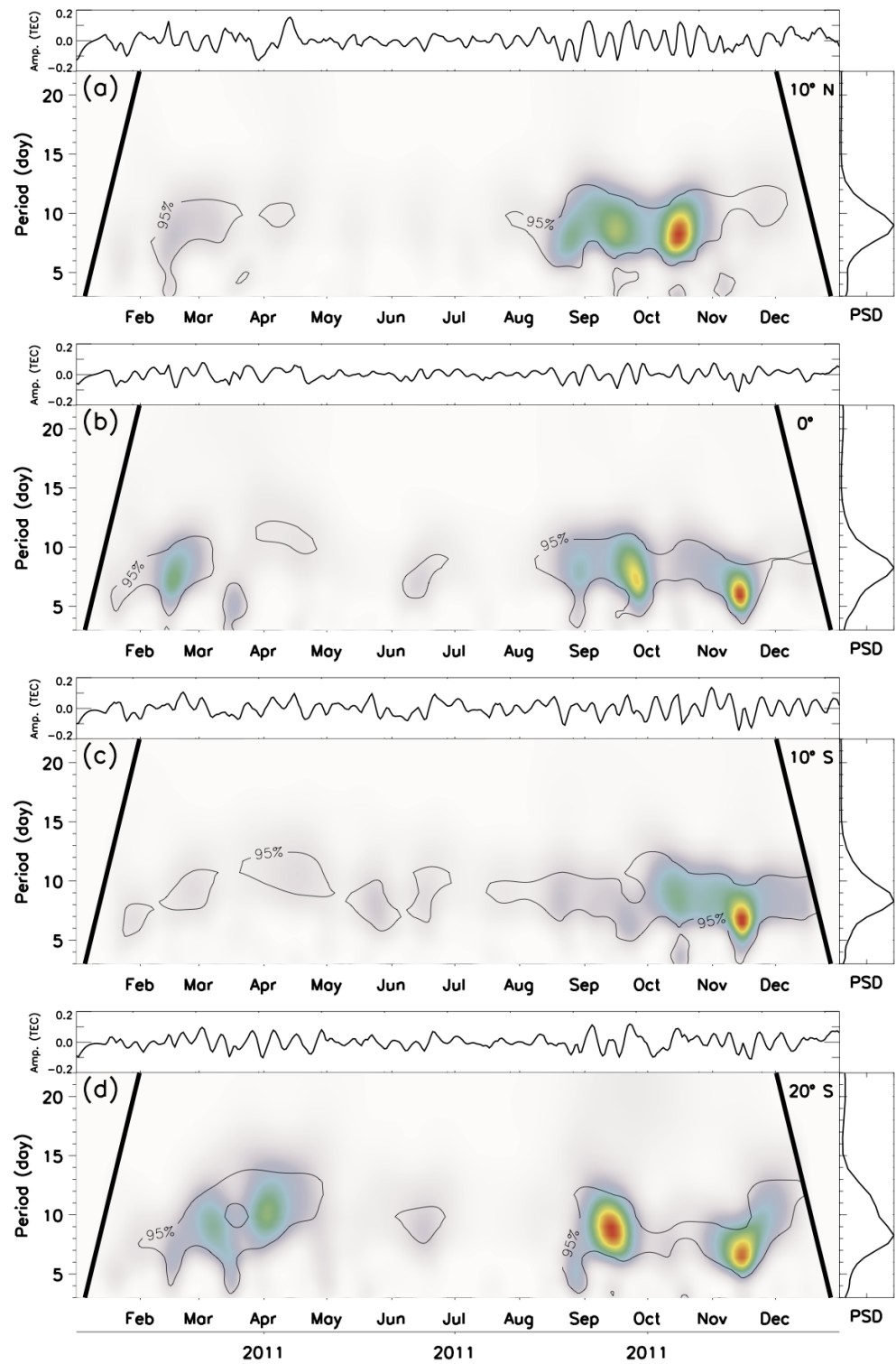

**Figure 6.** Same as Figure 3, but considering only periods shorter than 25 days during 2011.

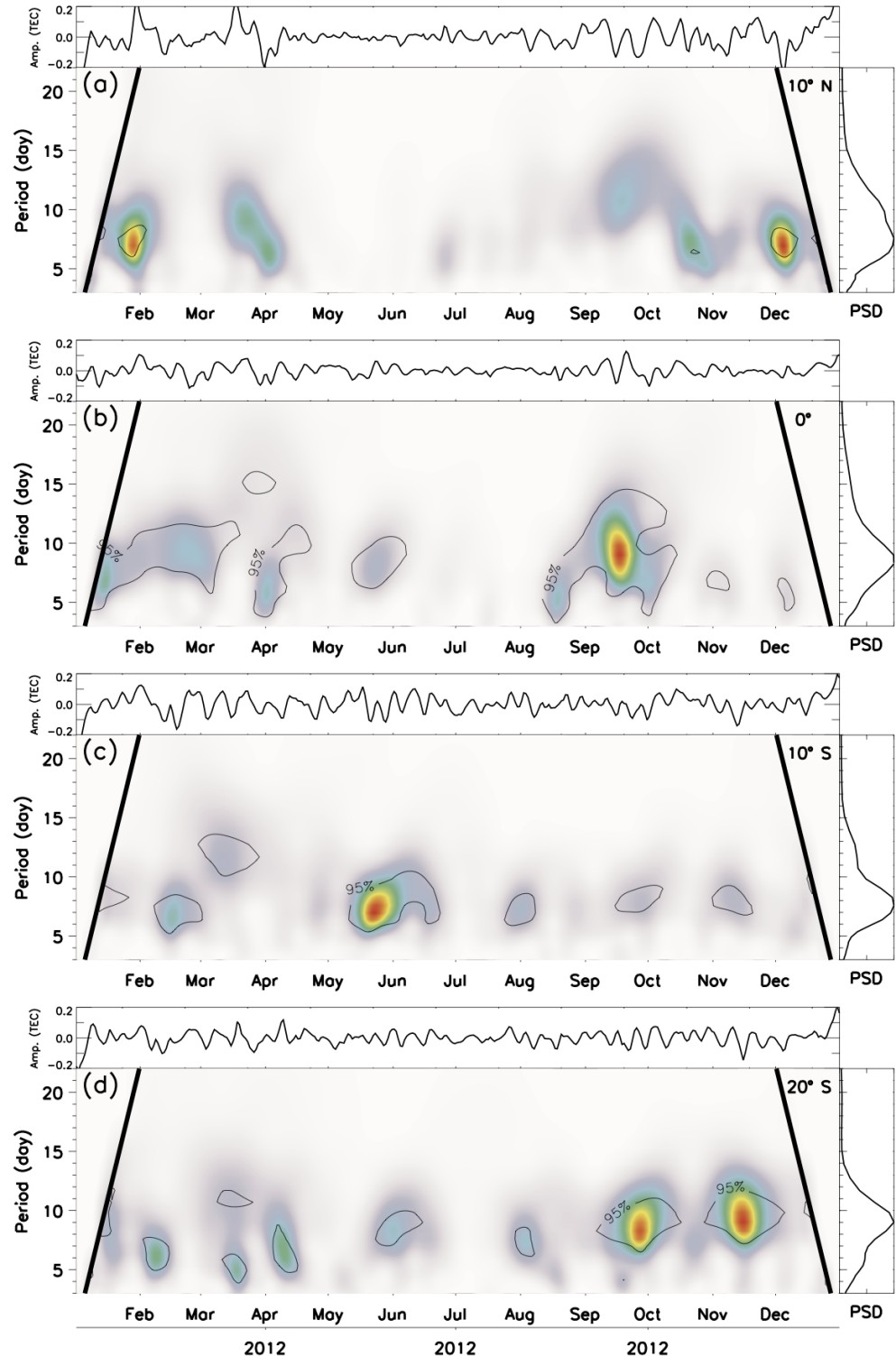

**Figure 7.** Same as Figure 5 for 2012.

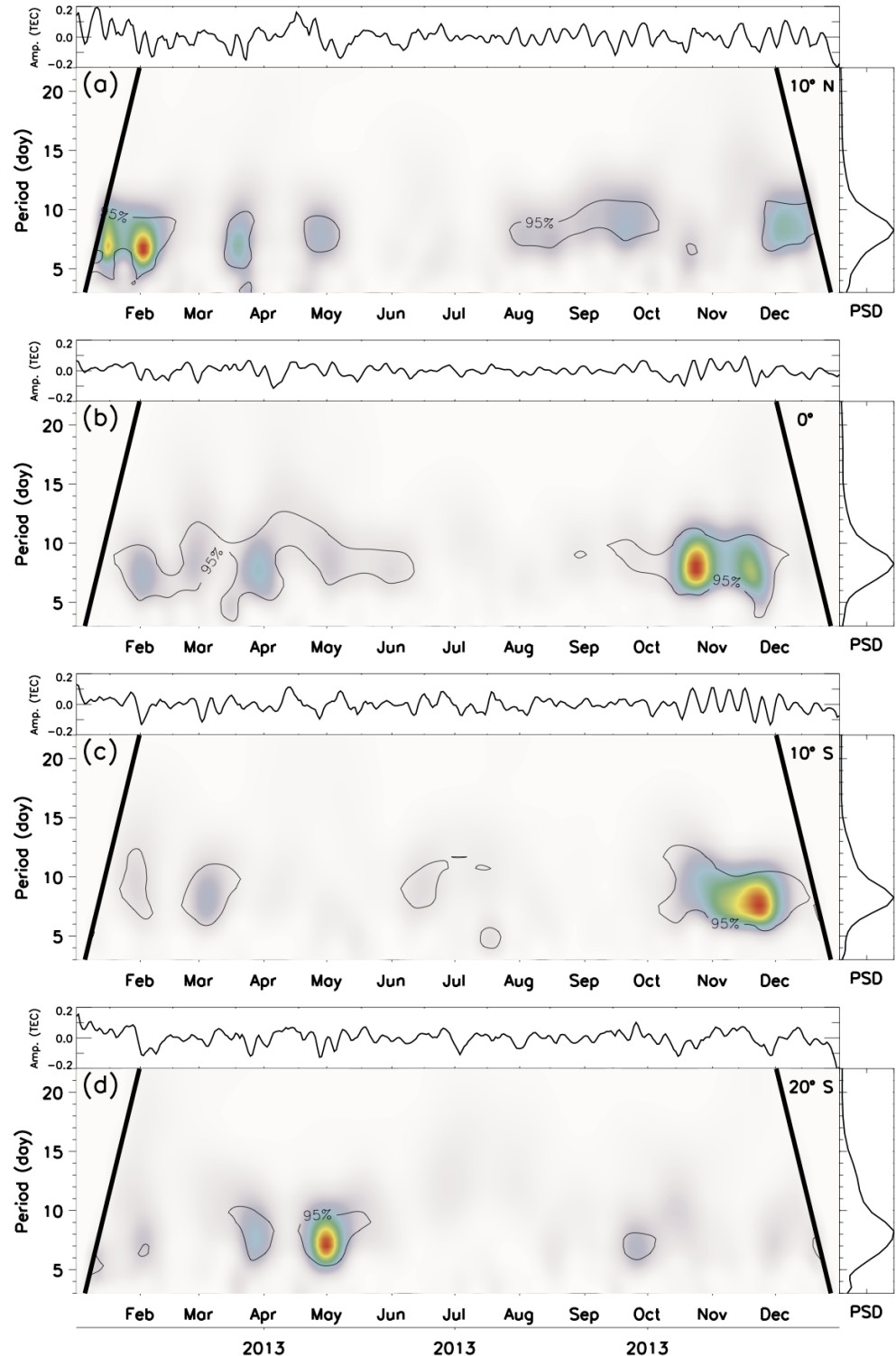

**Figure 8.** Same as Figure 5 for 2013.

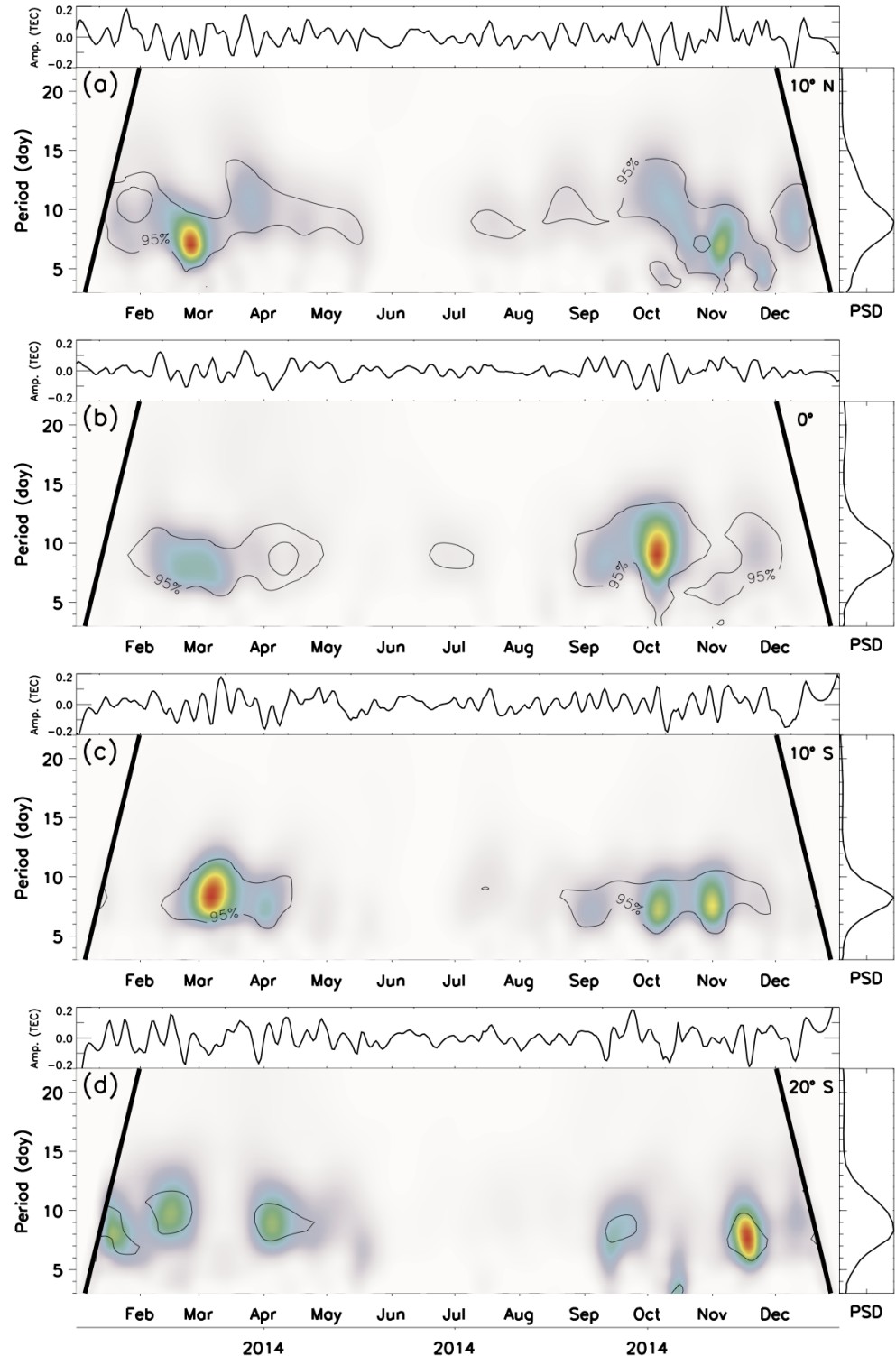

**Figure 9.** Same as Figure 5 for 2014.

## 3 Discussion and summary

It is well known that the lunar semidiurnal tide has a predictable source. Thus, short and long variations observed in the amplitudes must reveal changes in the atmosphere where this tidal component is propagating. For instance, annual and semiannual variation in the amplitudes of the lunar tide have also been observed in the mesosphere and lower thermosphere (MLT) neutral wind (Paulino et al., 2015). Terannual variations have been observed and simulated in some atmospheric fields as well (e.g., Pedatella et al., 2012; Pedatella, 2014). However, more investigation is necessary to the better understanding the reason for those variability.

The results from Figures 3 and 4 show that the annual variation is always present in the amplitudes of the lunar semidiurnal tide in the TEC. At the magnetic equator, the PSD of the annual variation is comparable to the semiannual, for instance. However, far from the equator, the annual variation is stronger. Furthermore, it seems that the annual variation is out of phase at this latitude compared to the annual variation observed in MLT winds (Figure 3, bottom row of Paulino et al., 2015), i.e., the annual variation maximizes around January for all latitudes in the TEC and it maximizes around November in the MLT winds. This reinforces that the lunar tide obeys the changes in the atmosphere and the observed variability is not due to changes in the sources.

Although the semiannual oscillation raised up as the second peak in the Lomb-Scargle Periodogram (Figure 3) from 2011 to 2014, it appeared sporadically and with more intensity in lower latitudes. In contrast, the TEC observed in Brazil shows a semiannual variation and have maxima around the equinox during both low and high solar activities (Jonah et al., 2015).

The terannual variation with period around 120 days, in average, was the third peak found in the amplitudes of the semidiurnal tide. It was sporadic at $10^o$ N, $0^o$ and $20^o$ S. At $10^o$ S, it was present in almost the whole period of observation and it was stronger than the semiannual oscillation during the first two years of observations, except at $20^o$S. Oscillations with 70-80 days period were also observed in all latitudes sporadically, mainly in the beginning of 2013 and 2014. In Paulino et al. (2017), it is possible to see that the terannual oscillation appears evident at magnetic latitudes out of the equator. It is probable that the combination of the annual variation with maximum in the austral summer months and semidiurnal variation with maximums around the equinoxes (matching with the TEC maximums) is producing the terannual variation in the amplitudes of the lunar semidiurnal tide.

Figure 1(a) shows roughly a short period oscillation in the amplitudes of the lunar tide which can be observed in all studied latitudes over Brazil. Sometimes, this short period oscillation is stronger and sometimes it is very tenuous. This bahavior is quite interesting because it was also observed before (see Figure 3 of Paulino et al., 2017).

An interesting aspect reveled in this work was the periodicity close to 8 days. Based on the literature, there are two kinds of large scale waves with periods close to 8 days: (1) Fast Kelvin wave (e.g., Abdu et al., 2015, and references therein) and (2) Quasi 10 days planetary wave (e.g., Forbes and Zhang, 2015; Yamazaki and Matthias, 2019).

Fast Kelvin waves have been observed with period typically of 6-10 days. It is a kind of wave trapped in the equatorial region which has characteristics for gravity waves, i.e., it obeys the dispersion relation of gravity waves. Fast Kelvin waves are typically observed with large amplitude in the zonal wind component and insignificant amplitudes in the meridional one. As

the present Q8D oscillation was observed close to the equator as well as at 20$^o$ magnetic latitude, which can be out of the tropic in the west part of Brazil, this modulation can have contribution of other oscillations as well. Some observations have shown that the amplitudes of the Fast Kelvin wave dominate in altitudes below 90 km (e.g., Lieberman and Riggin, 1997). Dhanya et al. (2012) found periodicities close to 8 days in the Equatorial electrojet current and in the MLT winds and associated that oscillation to Fast Kelvin wave. Abdu et al. (2015) also observed Q8D oscillation in the vertical drift of F region which

modulated the spread-F development.

    Figure 10 shows the zonal (solid line) and meridional (dashed line) thermospheric mean wind at 00:00 UT measured by two Fabry-Perot interferometers deployed at Cajazeiras (6.9$^o$S, 38.5$^o$W) and São João Cariri (7.4$^o$S, 36.5$^o$W) during November 2013. Further details about the operations of those equipments can be found in Makela et al. (2009). One can observe that there is an almost in phase oscillation of about one week in the wind field maybe suggesting the same origin of the observed Q8D

oscillation in the amplitude of the lunar tide during this epoch.

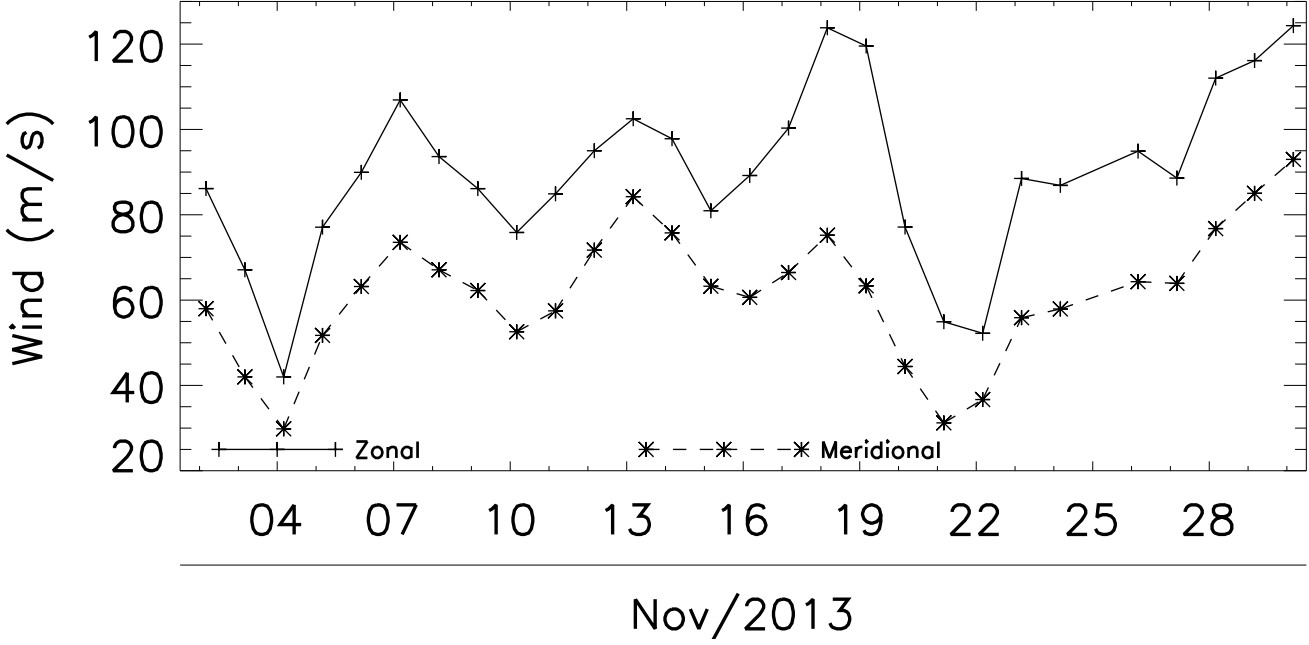

**Figure 10.** Zonal (solid line) and Meridional (dashed line) means wind components measured two Fabry-Perot interferomenters over Cajazeiras and São João do Cariri measured at 00:00 UT during November 2013.

    Figure 11 shows the Lomb-Scargle periodogram for those data. A strong oscillation of $\sim 6$ days can be observed and it is likely associated with the presence of fast Kelvin wave in the equatorial zone. One can also observe a peak of quasi 10 days in both components, however, it was below the confidence level.

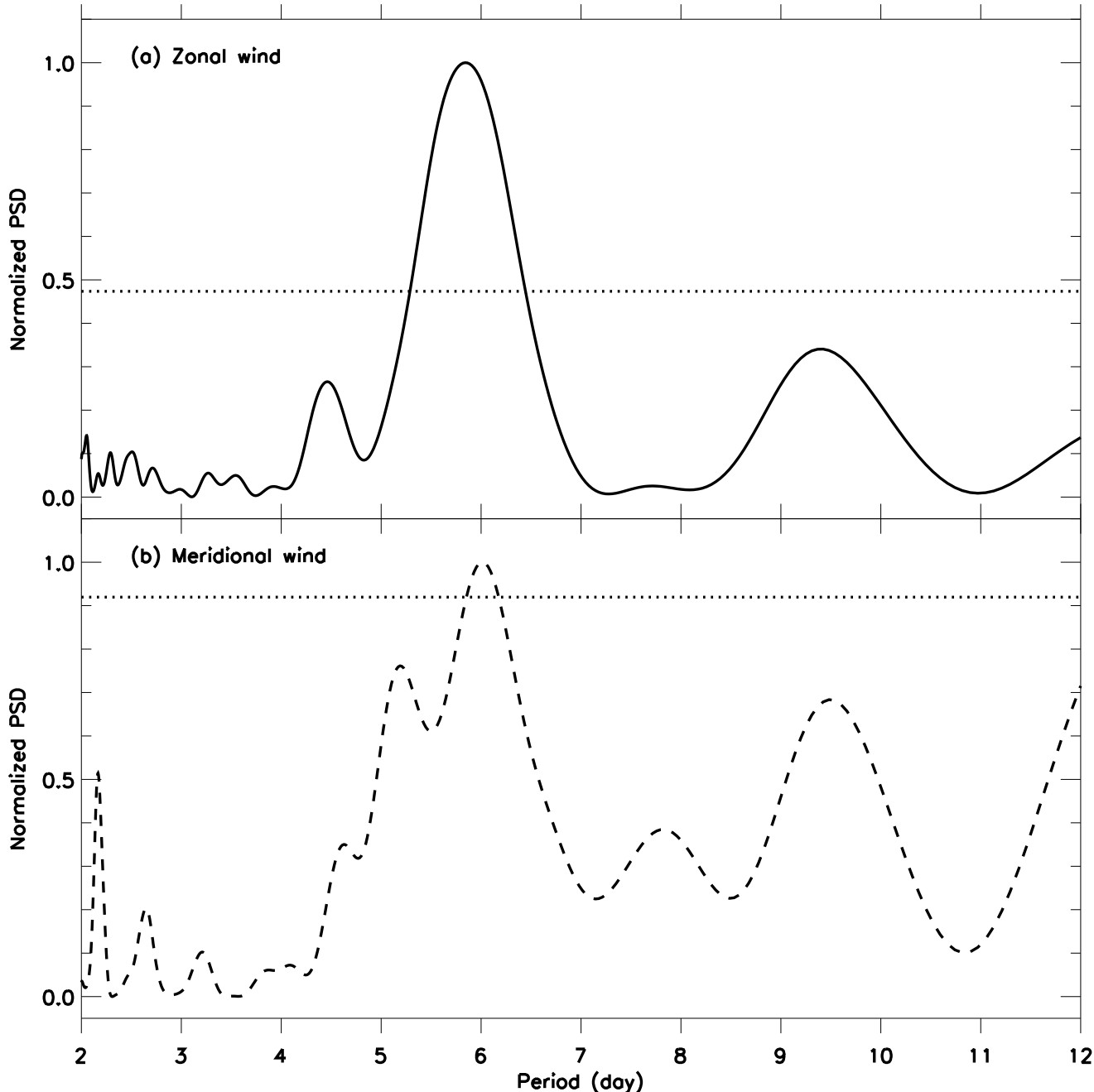

**Figure 11.** Lomb-Scargle periodogram for (a) zonal and (b) meridional wind during November 2013 in the thermosphere over the equatorial region. Horizontal dotted line represents a significancy level of 95%, i.e., false alarm probability of 0.05.

In the past years, the interest in studying the quasi 10 day wave (Q10DW) has been recovered, primarily due to its association with polar Sudden Stratospheric Warmings (SSWs) (Yamazaki and Matthias, 2019; Mo and Zhang, 2020). Another motivation

was the long term observation from satellites that allow to investigate seasonality, year to year variation and spatial (latitude x longitude x altitude) dependencies (Forbes and Zhang, 2015; John and Kumar, 2016).

Although the present results concentrate into the oscillations around 8 days in the amplitudes of the semidiurnal lunar tide, these oscillations have some characteristics similar to the Q10DWs as pointed out by Forbes and Zhang (2015). For instance, in some years, they have large amplitudes during the equinox and winter months in both hemispheres.

Observation of wave/oscillation with periods 8-10 days have been made in the thermosphere-ionosphere. For example, Forbes (1996) found Q10DW oscillations in the mesopause and low thermosphere region using data of Medium Frequency (MF) radar and a magnetometer. Pancheva and Laštovička (1998) observed fluctuations of 7-8 days from November to December 1994 during an international Campaign. Abdu et al. (2006) studied variation in the Equatorial electrojet (EEJ) current and in the MLT wind and showed the presence of 8-12 days oscillation around the equinox of 1999 in the equatorial region. Jacobi et al. (2007) observed 7-12 days waves in the TEC maps over Europe region. Jonah et al. (2015) also observed 8-10 days oscillation in TEC over Brazil, primarily around the equinoxes. More recently, comprehensive studies on Q10DW using satellite data presented some important temporal and spatial characteristics of this wave below 110 km altitude (Forbes and Zhang, 2015; John and Kumar, 2016). Additionally, Yamazaki and Matthias (2019) and Mo and Zhang (2020) presented results associating the Q10W to Sudden Stratospheric Warming events.

It is important to note that the observations above showed periods varying from 8 to 10 days in both the mesosphere and thermosphere-ionosphere. Forbes and Zhang (2015) showed slight variation (0.4 days) in the period of Q10W (9.7 to 9.9 days) and concluded that the doppler shift produced by the horizontal wind can change the period of the wave. The present study uses amplitudes of the lunar semidiurnal tide in lunar time, i.e., the lunar days is about 0.036 days longer than solar day. Assuming that the observed oscillation has a period of 8.5 days in lunar time, it corresponds to 8.806 days in solar time. However, this is not enough to explain the discrepancy between the observed period of the Q10DW in the lower atmosphere and the present results.

Figure 12 shows the horizontal wind at 93 km altitude over Cachoeira Paulista ($22.7^0$S; $45.0^0$W) during November 2013 measured at 02:00 Universal Time. Further details about the wind measurements using meteor radar have been published elsewhere (e.g., Paulino et al., 2012). The temporal evolution of the winds shows some periodic oscillations.

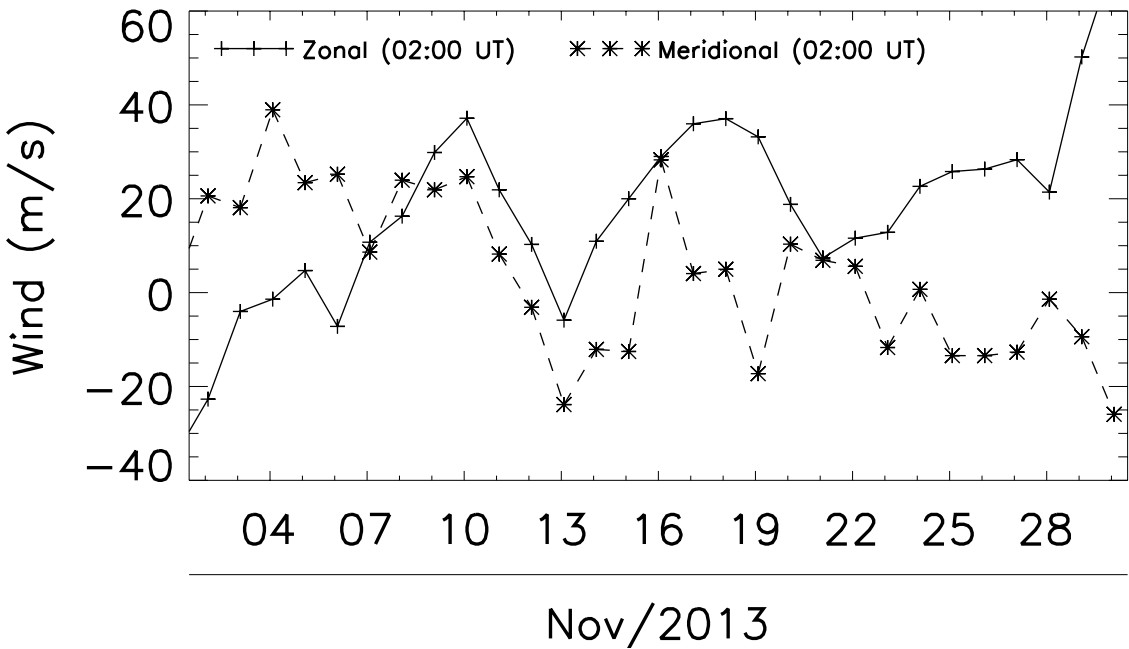

**Figure 12.** Zonal (solid line) and Meridional (dashed line) wind components at 93 km height over Cachoeira Paulista measured at 02:00 UT during November 2013.

Figure 13 shows the Lomb-Scargle periodogram of the wind including all temporal measurements at 93 km. Quasi 10 days oscillation is shown in both zonal and meridional components of the horizontal wind. In the zonal component the peak of the oscillation was concentrated at 9.7 day, while in the meridional one, the peak was at ∼9 days. The presence of this simultaneous oscillation in the MLT wind strongly suggests that the observed oscillation in amplitude of the lunar tide during November 2013 220 could have contribution of this oscillation, at least, out of the equator. But the coupling mechanism needs further investigations.

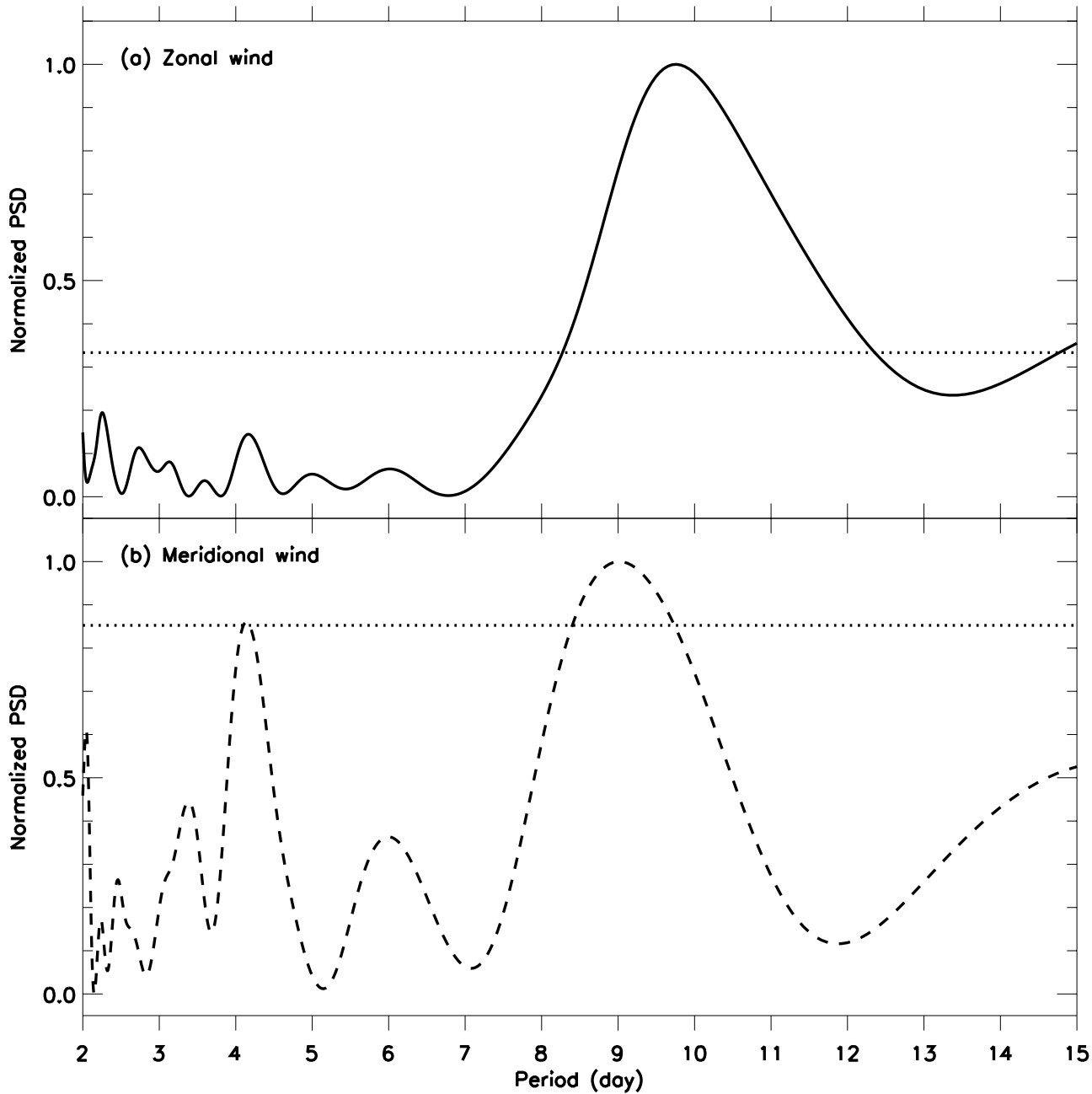

**Figure 13.** Lomb-Scargle periodogram for (a) zonal and (b) meridional wind during November 2013 at 93 km altitude over Cachoeira Paulista. Horizontal dotted line represents a significancy level of 95%, i.e., false alarm probability of 0.05.

The main results of this investigation can be summarized as follows:

- There is a strong temporal variability in the amplitudes of the lunar semidiurnal tides calculated in the TEC maps over Brazil from 2011 to 2014;

- Annual variation in the lunar semidiurnal tide is always present in all observed latitudes and it is dominant in lower latitudes;

- Semiannual and terannual ($\sim$120 days) were, respectively, the second and third most important long period oscillation observed in the amplitudes of the lunar tide. However, it was observed a temporal and spatial variability of these oscillations, which allow them to become dominant in a given time interval and latitude range;

- The observed dominant oscillations in the amplitudes of the lunar semidiurnal tide had periods between 8-11 days, with maxima around the equinoxes. In some years, as 2013 and 2014, the peaks occurred in the winter;

- Coincident measurements of the horizontal wind during November 2013 show the presence quasi 10 days oscillation in MLT at low latitudes ($23^o$S) and quasi 6 days oscillation in the equatorial thermosphere.

Based on the present main results for the short period oscillations in the amplitudes of the lunar semidiurnal tide, i.e., the Q8D oscillation, it is clear that there is discrepancy between the observed Q8D and other oscillations observed in the wind. Additionally, according to the theory of planetary waves, the dissipative process into the thermosphere does not allow direct propagation of these wave to high levels. Then, the explanation for observation of planetary waves in the thermosphere-ionosphere have been suggested, basically, based on two possibilities: (1) modulation of the tidal amplitudes, in special, the semidiurnal components which can propagate to higher altitudes into the thermosphere and/or (2) through the theory of dynamo on the electrodynamics of the ionosphere. The present results suggest that maybe a combination of these two possibiliteis could be necessary in other to explain the observed modulation in the amplitude of the lunar semidiurnal tide. However, further investigations are necessary to understand this coupling mechanism.

*Data availability.* The TEC maps used in this work are available online in the EMBRACE website (http://www2.inpe.br/ climaespacial). Meteor winds can be requested to Dr. Lourivaldo Mota Lima (lourivaldo_mota@yahoo.com.br). Thermospheric winds for São João do Cariri and Cajazeiras can be dowloaded from Madrigal CEDAR data base.

*Author contributions.* ARP has written the manuscript and performed the analysis in the data base. FSA has worked in the Lomb-Scargle periodograms. IP has revised the manuscript and helped in some analysis of the data. CMW has provided the TEC maps. LML has provided the meteor winds. PPB is responsilbe for the meteor winds and has revised the manuscript. ISB has revised the manuscript.

*Competing interests.* The authors declare that they do not have competing interests

*Acknowledgements.* A. R. Paulino thanks to Coordenação de Aperfeiçoamento de Pessonal de Nível Superior (CAPES) for the scholarship. A. R. Paulino, I. Paulino, C, M. Wrasse and I. S. Batista thank to Conselho Nacional de Desenvolvimento Científico e Tecnolóligo (CNPq) for the financial support under contracts #460624/2014-8, #303511/2017-6, #307653/2017-0, #405555/2018-0 and #306844/2019-2. A. R. Paulino and I. Paulino thank to the Fundação de Amparo à Pesquisa do Estado da Paraíba by the PRONEX grant (002/2019). Wavelet software was provided by C. Torrence and G. Compo and it is available at: http://paos.colorado.edu/research/wavelets/. The authors thank to R. A. Buriti, J. J. Makela and J. W. Meriwether for kindly providing the FPI data.

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
