# Peer review of "Variability of the lunar semidiurnal tidal amplitudes in the ionosphere over Brazil"

_Annales Geophysicae, 2020_

## Referee Comment (RC1) · Anonymous Referee #1 · 29 Jun 2020

**Review of ANGEO Manuscript: 2020-34**

**Variability of the lunar semidiurnal tidal amplitudes in the ionosphere over Brazil**
Ana Roberta Paulino, Fabiano da Silva Araújo, Igo Paulino, Cristiano Max Wrasse,
Lourivaldo Mota Lima, Paulo Prado Batista, and Inez Staciarini Batista

The variability in the amplitudes of the lunar semidiurnal tide is investigated using TEC maps over Brazil from January 2011 to December 2014. The authors find evidence of strong annual variation. Semiannual and intra-seasonal oscillations are found to be the second and third largest components, respectively. Among the short-period oscillations in the amplitude of the lunar tide, the most pronounced ones were concentrated between 7-11 days, which the authors ascribe to the normal mode westward propagating quasi 10 days planetary wave with horizontal wavenumber equal to 1. The presented results suggest a possible coupling process by modulation of the lunar semidiurnal tidal amplitudes that allows the propagation of the 7-11 days waves into the thermosphere-ionosphere system.

**General comment**

While the manuscript contains some interesting results, I cannot recommend publication in the present form for the following two reasons:

(1) The language should be improved.
(2) Additional observational and/or modeling work is needed to demonstrate that the ~9-day oscillation is indeed consistent with a westward quasi-10-day normal mode.

**Other comments and technical corrections**

- Line 4 (and throughout): Intra-seasonal variability is usually referred to variations less than ~90 days.
- Line 6: in special → in particular
- Lines 9-10 (see comment before): This sentence and result is highly speculative. More modeling or observational work should done to demonstrate a link to the 10-day normal mode.
- Lines 19-20: need reference
- Line 22: Ozone → ozone
- Lines 26-27: these ranges are not consistent with other studies. Need reference.
- Lines 48-49: define what is meant with long and short period
- Section 2: More details on how the lunar tide is calculated are needed.
- Line 60: reference for the filter needed
- Line 77-78 and Figure 3: need to discuss the 70- to 80-day variations and 120 variations
- Wavelet plots: need to include a confidence level
- Line 91: 30 days or 25 days as reported in the legend of Figure 4?
- Line 131: maiximuma → maxima
- Lines 133-134: the oscillation at 70-80 days is almost at large. Need to comment on this.

- Lines 144-148: this statement is highly speculative. Need additional modeling work or analysis of concurrent observations at different longitudinal locations.
- Line 161: thermosphere misspelled
- Line 171: to notice → to note
- Lines 192-194: can the observed intra-seasonal variability be related to Madden-Julian Oscillation?
- Lines 197-198: need to further elaborate this point.
- Line 201: preset → present
- Line 208: though → through
- Lines 209-2010: analysis insufficient to support this statement

---

## Referee Comment (RC2) · Anonymous Referee #2 · 8 Jul 2020

The authors examined seasonal and intraseasonal variability of the semidiurnal lunar tide in TEC over Brazil. The main finding is that the amplitude of the semidiurnal lunar tide in TEC often shows 7-11 day variations. The authors speculate that these variations are associated with the quasi-10-day wave in the middle atmosphere.

Although the results are interesting, I am not totally convinced that the authors were able to extract 7-11 day variations of the semidiurnal lunar tide. The authors used the technique of Paulino et al. (2017) to derive the amplitude of the semidiurnal lunar tide. This technique involves a 27 day window, which enables to distinguish between the semidiurnal lunar tide (12.42h) and semidiurnal solar tide (12.00h). The technique should largely eliminate variations with periods less than 27 days, even though the amplitude is calculated for each day. Thus, it is unclear whether the presented short-

period variations are meaningful. The authors are advised to check the spectrum of the original TEC data (instead of the spectrum of the semidiurnal lunar tide) to confirm that a spectral peak exists at the semidiurnal lunar tide (12.42h) as well as the sideband frequency corresponding to the quasi-10-day wave modulation of the semidiurnal lunar tide.

Other comments:

1. Equation (1)

This needs more explanations. What is "filter" on the left-hand side? How is it applied to the data?

2. Lomb-Scargle periodogram

Since the authors show wavelet spectra in Figures 3 and 5-8, Lomb-Scargle periodograms in Figures 2 and 4 do not seem necessary. I suggest to remove them.

3. Figure 9

The antisymmetric mode such as the quasi-10-day wave has the phase structure that is antisymmetric about the equator, but not the amplitude structure. That is, when there is a strong quasi-10-day wave, we should expect the amplitude of the wave to be large in both northern and southern hemispheres but with the opposite phase. What is shown in Figure 9 is the anti-correlation of the amplitude between the northern and southern hemispheres, which does not necessarily support the involvement of the quasi-10-day wave.

---

## Author Comment (AC1) · 5 Aug 2020

REVIEWER:"**The variability in the amplitudes of the lunar semidiurnal tide is investigated using TEC maps over Brazil from January 2011 to December 2014. The authors find evidence of strong annual variation. Semiannual and intra-seasonal oscillations are found to be the second and third largest components, respectively. Among the short-period oscillations in the amplitude of the lunar tide, the most pronounced ones were concentrated between 7-11 days, which the authors ascribe to the normal mode westward propagating quasi 10 days planetary wave with horizontal wavenumber equal to 1. The presented results suggest a possible coupling process by modulation of the lunar semidiurnal tidal amplitudes that allows the propagation of the 7-11 days waves into the thermosphere-**

**ionosphere system.**"

AUTHORS: We are grateful for the dedicated time in reading and suggesting improvements to our paper.

REVIEWER:"**While the manuscript contains some interesting results, I cannot recommend publication in the present form for the following two reasons: (1) The language should be improved. (2) Additional observational and/or modeling work is needed to demonstrate that the ∼9-day oscillation is indeed consistent with a westward quasi-10-day normal mode.**"

AUTHORS: The reviewer is right that the results are not conclusive at all. We have revised the manuscript excluding the non-conclusive results. We have tried to investigate the horizontal propagation of the wave using data from stations separated by ∼ 3,600 km, but the results were not conclusive as well. Maybe we need use receivers located in Brazil, Africa, India and Indonesia to be reach confident results. We understand that this is out of the scope of this manuscript and we will not have enough time to do this kind of analysis within the schedule of publication of ANGEO. Therefore, we kindly ask to the reviewer to check whether the modification implemented in the manuscript can address his/her concerns. We have also revised the language according to the suggestions from the reviewer.

REVIEWER:

"**Line 4 (and throughout): Intra-seasonal variability is usually referred to variations less than 90 days.**"

"**Line 6: in special − > in particular**"

"**Lines 19-20: need reference**"

"**Line 22: Ozone** $->$ **ozone**"

"**Lines 26-27: these ranges are not consistent with other studies. Need reference.**"

"**Lines 48-49: define what is meant with long and short period.**"

"**Line 60: reference for the filter needed.**"

"**Line 91: 30 days or 25 days as reported in the legend of Figure 4?**"

"**Line 131: maiximuma** $->$ **maxima.**"

"**Line 161: thermosphere misspelled.**"

"**Line 171: to notice** $->$ **to note.**"

"**Line 201: preset** $->$ **present.**"

"**Line 208: though** $->$ **through.**"

AUTHORS: All those minor points have been corrected according to the suggestions.

REVIEWER: **"Lines 9-10 (see comment before): This sentence and result is highly speculative. More modeling or observational work should done to demonstrate a link to the 10- day normal mode."**

AUTHORS: We agree with the reviewer and we have revised it.

REVIEWER: **"Section 2: More details on how the lunar tide is calculated are needed."**

AUTHORS: We have added a subsection to explain the determination of the lunar tide as suggested.

REVIEWER: **"Line 77-78 and Figure 3: need to discuss the 70- to 80-day variations and 120 variations."**

AUTHORS: We have added the description of these oscillations as mentioned.

REVIEWER: **"Wavelet plots: need to include a confidence level."**

AUTHORS: We have included it. Thank you for the suggestion!

REVIEWER: **"Lines 133-134: the oscillation at 70-80 days is almost at large. Need to comment on this."**

AUTHORS: Thank you. We have commented it.

REVIEWER: **"Lines 144-148: this statement is highly speculative. Need additional modeling work or analysis of concurrent observations at different longitudinal locations."**

AUTHORS: We have also revised this statement. Thank you for this contribution.

REVIEWER: **"Lines 192-194: can the observed intra-seasonal variability be related to Madden-Julian Oscillation?"**

AUTHORS: It is very difficult correlate these events without further analysis.

REVIEWER: **"Lines 197-198: need to further elaborate this point."**

AUTHORS: We have removed the analysis about the symmetry of these oscillations

because it was discussed considering the magnetic latitudes instead of geographic ones. We are not sure how is the behavior of the symmetry of the planetary waves using magnetic coordinates, primarily in Brazil where there is a strong magnetic declination.

REVIEWER: **"Lines 209-2010: analysis insufficient to support this statement."**

AUTHORS: Yes, we have revised it.

Please also note the supplement to this comment:
https://angeo.copernicus.org/preprints/angeo-2020-34/angeo-2020-34-AC1-supplement.pdf

**Supplement:**

**Variability of the lunar semidiurnal tidal amplitudes in the ionosphere over Brazil**

Ana Roberta Paulino[1], Fabiano da Silva Araújo[1], Igo Paulino[2], Cristiano Max Wrasse[3], Lourivaldo Mota Lima[1], Paulo Prado Batista[3], and Inez Staciarini Batista[3]

[1]Departamento de Física, Universidade Estadual da Paraíba, Campina Grande, Brazil
[2]Unidade Acadêmica de Física, Universidade Federal de Campina Grande, Campina Grande, Brazil
[3]Divisão de Aeronomia, Instituto Nacional de Pesquisas Espaciais, São José dos Campos, Brazil

**Correspondence:** Ana Roberta Paulino (arspaulino@gmail.com)

**Abstract.** The variability in the amplitudes of the lunar semidiurnal tide was investigated using maps of Total Electron Content over Brazil from January 2011 to December 2014. Long period variability showed that the annual variation is always present in all investigated magnetic latitudes and it represents the main component of the temporal variability. Semiannual and  terannual (∼120 days) oscillations were the second and third components, respectively, but they presented significant temporal and spatial variation without a well-defined pattern. Among the short period oscillations in the amplitude of the lunar tide, the most pronounced ones were concentrated between 7-11 days. These oscillations were stronger around the equinoxes, in  particular between September and November in almost all latitudes. In some years, as in 2013 and 2014, for instance, they appeared with large power spectral density in the winter hemisphere.  In addition, using data from a meteor radar located at low latitudes in Brazil for November 2013, when the amplitude of the 7-11 days oscillation was strong, it was possible to identify the presence of quasi 10 days oscillation in the both zonal and meridional component of the horizontal winds.  However, equatorial fast Kelvin could be another possibility for the dynamical coupling of the thermosphere-ionosphere system that are resulting these observed oscillations in the amplitudes of the lunar tide.

[revised manuscript text omitted]

320   physical Research: Atmospheres, 124, 9874–9892, https://doi.org/10.1029/2019JD030634, 2019.

---

## Author Comment (AC2) · 5 Aug 2020

REVIEWER:"**The authors examined seasonal and intraseasonal variability of the semidiurnal lunar tide in TEC over Brazil. The main finding is that the amplitude of the semidiurnal lunar tide in TEC often shows 7-11 day variations. The authors speculate that these variations are associated with the quasi-10-day wave in the middle atmosphere.**"

AUTHORS: We appreciate the revision and the contributions from the Reviewer # 2. We have done our best to address all of the concerns from the reviewer.

REVIEWER:"**Although the results are interesting, I am not totally convinced that**

**the authors were able to extract 7-11 day variations of the semidiurnal lunar tide. The authors used the technique of Paulino et al. (2017) to derive the amplitude of the semidiurnal lunar tide. This technique involves a 27 day window, which enables to distinguish between the semidiurnal lunar tide (12.42h) and semidi- urnal solar tide (12.00h). The technique should largely eliminate variations with periods less than 27 days, even though the amplitude is calculated for each day. Thus, it is unclear whether the presented short- period variations are meaning- ful."**

AUTHORS: We thank the reviewer for this important comment. The TEC in the tropical region is mainly produced by the absorption of the EUV and X-rays solar radiations. Thus the diurnal cycle is faraway dominant and should be removed. The determination of the lunar semidiurnal tides in TEC maps were done according to the Pedatella and Forbes (2010) methodology and only quiet days were considered ($K_p < 3$) in the anal- ysis. Figure 1 (upper panel) shows a 29.5 days window of TEC from 27 July 2011 to 25 August 2011 measured at ($15^o$S, $39^o$W).

After the elimination of geomagnetic influences, a Fourier analysis was performed to extract the subharmonics of the solar day (diurnal, semidiurnal and terdiurnal oscil- lations). Effects of the solar rotation was removed using a 27-day window moving it forward one day at time to calculate the mean solar day centered in the window as can be seen in Figure 1 (middle panel). In addition, residual TEC was determined subtract the original TEC from the recovered one. Figure 1 (bottom panel) shows the residual TEC for this example, where the power of the diurnal cycle is reduced and other oscilla- tions can be observed. In the relative residual data (residual TEC divided by the mean TEC), a least square analysis in a window of 29-day was applied using the following equation:

$$y(\tau) = \sum_{n=0}^{2} A_n \cos\left(n\tau + \phi_n\right) \qquad (1)$$

where $\tau$ if the lunar time given by $\tau = t - \nu$, $\nu$ in the age of the Moon, which is set to be $0$ at the New Moon. The solar time is represented by $t$, the amplitudes and phases of the lunar tide components are represented by $A_n$ and $\phi_n$, respectively.

We have included this explanation in the manuscript in order to clarify the methodology. It was requested by Reviewer #1 as well. Furthermore, the scope of the present work was investigate the day-to-day variability of the amplitude of the lunar semidiurnal tide, which was calculated for each day and change as well as shown in Pedatella and Forbes (2010) and Paulino et al. (2017). Additionally, the determination of the short-period oscillations were statistically significant as in the LS periodogram as in the wavelet analysis (the significance level were included in all plots).

REVIEWER:" **The authors are advised to check the spectrum of the original TEC data (instead of the spectrum of the semidiurnal lunar tide) to confirm that a spectral peak exists at the semidiurnal lunar tide (12.42h) as well as the sideband frequency corresponding to the quasi-10-day wave modulation of the semidiurnal lunar tide.**"

AUTHORS: We thank to the reviewer for this comments. Regarding to the presence of the lunar tide in the TEC, Figure 1 of Paulino et al. (2017) shows very clear evidences in different periods.

On the other hand, we have followed the suggestion from Reviewer #2 to show the presence of the Q8D wave in the TEC. Figure 2(a) shows the data of the quiet day TEC maps for November 2013 (when the Q8D wave was strong in the amplitude of the semidiurnal lunar tide) at $8^o$S, $35^o$W (where there are confident GNSS receivers and the

amplitude of the amplitude of the Q8D was strong). One can see that there is a strong day-to-day variability in the TEC. Figure 1(b) shows the Lomb-Scargle periodogram calculated using the data from Figure 1 (a). The diurnal cycle is very pronounced compared to the other oscillation. Figure 1(c) shows the same results of Figure 1(b) but ranged from 0 to 50 PSD units. One can see that the Q8D wave peak is above of the confidence level and it demonstrates what was suggested by the reviewer.

REVIEWER:"**1. Equation (1) - This needs more explanations. What is "filter" on the left-hand side? How is it applied to the data?"**

AUTHORS: Thank you for the comment. We have included a citation about the application of the filter as suggested by the Reviewer #1. To apply the filter, first we apply the FFT transform, then we multiply the "filter" by the FFT signal. Finally we apply the inverse FFT to recover the filtered signal.

REVIEWER:"**Lomb-Scargle periodogram - Since the authors show wavelet spectra in Figures 3 and 5-8, Lomb-Scargle periodograms in Figures 2 and 4 do not seem necessary. I suggest to remove them.**"

AUTHORS: The reviewer is right! Most of the aspects showed in Figure 2 and 4 can be seen in the wavelet charts. However, LS periodograms can give us a general idea about the periodicities using the whole period of analysis and comparisons between the latitudes are easily matched. Even so, if the reviewer thinks better to remove them, we can do it for the revised version.

REVIEWER:"**Figure 9 - The antisymmetric mode such as the quasi-10-day wave has the phase structure that is antisymmetric about the equator, but not the amplitude structure. That is, when there is a strong quasi-10-day wave, we should**

**expect the amplitude of the wave to be large in both northern and southern hemispheres but with the opposite phase. What is shown in Figure 9 is the anti-correlation of the amplitude between the northern and southern hemispheres, which does not necessarily support the involvement of the quasi-10-day wave."**

AUTHORS: The reviewer is right! Figure 9 does not necessarily support the anti-symmetry. Thank you for the comment. Besides, we are not sure about how is the symmetry of planetary waves regarding the magnetic coordinates. Therefore, we decide to remove these analysis. Thank you for this contribution.

Please also note the supplement to this comment:
https://angeo.copernicus.org/preprints/angeo-2020-34/angeo-2020-34-AC2-supplement.pdf
* * *
[Figure]

**Fig. 1.** Original TEC from 27 July 2011 calculated at (15o S, 39o W). (Middle panel) recovered signal using sub-hamornics of the solar day within a 27 days window. (Bottom panel) Residual TEC.

**Fig. 2.** (a) TEC data calculated to November 2013 at (8o S, 35o W) (b). Lomb-Scargle period-gram for the data TEC data shown in panel (a). (c) Same as Figure (b), but for zoomed to the y-range from 0 to

[Figure]

**Supplement:**

**Variability of the lunar semidiurnal tidal amplitudes in the ionosphere over Brazil**

Ana Roberta Paulino[1], Fabiano da Silva Araújo[1], Igo Paulino[2], Cristiano Max Wrasse[3], Lourivaldo Mota Lima[1], Paulo Prado Batista[3], and Inez Staciarini Batista[3]

[1]Departamento de Física, Universidade Estadual da Paraíba, Campina Grande, Brazil
[2]Unidade Acadêmica de Física, Universidade Federal de Campina Grande, Campina Grande, Brazil
[3]Divisão de Aeronomia, Instituto Nacional de Pesquisas Espaciais, São José dos Campos, Brazil

**Correspondence:** Ana Roberta Paulino (arspaulino@gmail.com)

**Abstract.** The variability in the amplitudes of the lunar semidiurnal tide was investigated using maps of Total Electron Content over Brazil from January 2011 to December 2014. Long period variability showed that the annual variation is always present in all investigated magnetic latitudes and it represents the main component of the temporal variability. Semiannual and  terannual (∼120 days) oscillations were the second and third components, respectively, but they presented significant temporal and spatial variation without a well-defined pattern. Among the short period oscillations in the amplitude of the lunar tide, the most pronounced ones were concentrated between 7-11 days. These oscillations were stronger around the equinoxes, in  particular between September and November in almost all latitudes. In some years, as in 2013 and 2014, for instance, they appeared with large power spectral density in the winter hemisphere.  In addition, using data from a meteor radar located at low latitudes in Brazil for November 2013, when the amplitude of the 7-11 days oscillation was strong, it was possible to identify the presence of quasi 10 days oscillation in the both zonal and meridional component of the horizontal winds.  However, equatorial fast Kelvin could be another possibility for the dynamical coupling of the thermosphere-ionosphere system that are resulting these observed oscillations in the amplitudes of the lunar tide.

[revised manuscript text omitted]

320   physical Research: Atmospheres, 124, 9874–9892, https://doi.org/10.1029/2019JD030634, 2019.

---

## Author Response (AR1)

**Responses to Editor and Reviewer**

**General Comments:**

Dear Dr. Ana G. Elias.

First of all, thank you for consider our paper for revision. We also thank to the reviewers for the important insights presented during their revisions. We have done our best to address the concerns of both reviewers properly. Our point-by-point responses are listed as following and the necessary modifications were highlighted in the the end of this file.

**Reviewer #1:**

REVIEWER: **"The variability in the amplitudes of the lunar semidiurnal tide is investigated using TEC maps over Brazil from January 2011 to December 2014. The authors find evidence of strong annual variation. Semiannual and intra-seasonal oscillations are found to be the second and third largest components, respectively. Among the short-period oscillations in the amplitude of the lunar tide, the most pronounced ones were concentrated between 7-11 days, which the authors ascribe to the normal mode westward propagating quasi 10 days planetary wave with horizontal wavenumber equal to 1. The presented results suggest a possible coupling process by modulation of the lunar semidiurnal tidal amplitudes that allows the propagation of the 7-11 days waves into the thermosphere-ionosphere system."**

AUTHORS: We are grateful for the dedicated time in reading and suggesting improvements to our paper.

REVIEWER: **"While the manuscript contains some interesting results, I cannot recommend publication in the present form for the following two reasons: (1) The language should be improved. (2) Additional observational and/or modeling work is needed to demonstrate that the ∼9-day oscillation is indeed consistent with a westward quasi-10-day normal mode."**

AUTHORS: The reviewer is right that the results are not conclusive at all. We have revised the manuscript excluding the non-conclusive results. We have tried to investigate the horizontal propagation of the wave observed in the amplitude of the lunar semidiurnal tide using data from stations separated by ∼ 3,600 km, but the results were not conclusive as well. We have tried to use global TEC maps from CEDAR database to investigate the presence of 8.5 days directly in the TEC. As the amplitude of the Q8D oscillation is small compared to the other oscillations, the phases were not well resolved and we could not conclude about the propagation direction (eastward or westward). Maybe we need use specific receivers located in Brazil, Africa, India and Indonesia to be reach confident results, but we understand that this is out of the scope of this manuscript and we will not have enough time to do this kind of analysis within the schedule of publication of ANGEO.

Therefore, we kindly ask to the reviewer to check whether the modification implemented in the manuscript can address his/her concerns. In addition, we have included analysis of the wind data from FPIs. The results showed the presence of quasi 6 days oscillations which can be another possibility to explain the observed results. Even without

a conclusion about the real coupling mechanism that are producing the Q8D oscillation in the amplitudes of the lunar semidiurnal tide in the TEC, we have to agree that the results are new and important to the scientific community and certainly it will start a relevant debate on this topic.

We have also revised the language according to the suggestions from the reviewer.

REVIEWER:

"Line 4 (and throughout): Intra-seasonal variability is usually referred to variations less than  90 days."

"Line 6: in special $->$ in particular"

"Lines 19-20: need reference"

"Line 22: Ozone $->$ ozone"

"Lines 26-27: these ranges are not consistent with other studies. Need reference."

"Lines 48-49: define what is meant with long and short period."

"Line 60: reference for the filter needed."

"Line 91: 30 days or 25 days as reported in the legend of Figure 4?"

"Line 131: maiximuma $->$ maxima."

"Line 161: thermosphere misspelled."

"Line 171: to notice $->$ to note."

"Line 201: preset $->$ present."

"Line 208: though $->$ through."

AUTHORS: All those minor points have been corrected according to the suggestions.

REVIEWER: "Lines 9-10 (see comment before): This sentence and result is highly speculative. More modeling or observational work should done to demonstrate a link to the 10- day normal mode."

AUTHORS: We agree with the reviewer and we have revised it.

REVIEWER: "Section 2: More details on how the lunar tide is calculated are needed."

AUTHORS: We have added a subsection to explain the determination of the lunar tide as suggested.

REVIEWER: "Line 77-78 and Figure 3: need to discuss the 70- to 80-day variations and 120 variations."

AUTHORS: We have added the description of these oscillations as mentioned.

REVIEWER: "Wavelet plots: need to include a confidence level."

AUTHORS: We have included it. Thank you for the suggestion!

REVIEWER: "Lines 133-134: the oscillation at 70-80 days is almost at large. Need to comment on this."

AUTHORS: Thank you. We have commented it.

REVIEWER: **"Lines 144-148: this statement is highly speculative. Need additional modeling work or analysis of concurrent observations at different longitudinal locations."**

AUTHORS: We have also revised this statement. Thank you for this contribution.

REVIEWER: **"Lines 192-194: can the observed intra-seasonal variability be related to Madden-Julian Oscillation?"**

AUTHORS: It is very difficult correlate these events without further analysis.

REVIEWER: **"Lines 197-198: need to further elaborate this point."**

AUTHORS: We have removed the analysis about the symmetry of these oscillations because it was discussed considering the magnetic latitudes instead of geographic ones. We are not sure how is the behavior of the symmetry of the planetary waves using magnetic coordinates, primarily in Brazil where there is a strong magnetic declination.

REVIEWER: **"Lines 209-2010: analysis insufficient to support this statement."**

AUTHORS: Yes, we have revised it.

**Reviewer #2:**

REVIEWER: **"The authors examined seasonal and intraseasonal variability of the semidiurnal lunar tide in TEC over Brazil. The main finding is that the amplitude of the semidiurnal lunar tide in TEC often shows 7-11 day variations. The authors speculate that these variations are associated with the quasi-10-day wave in the middle atmosphere."**

AUTHORS: We appreciate the revision and the contributions from the Reviewer # 2. We have done our best to address all of the concerns from the reviewer.

REVIEWER: **"Although the results are interesting, I am not totally convinced that the authors were able to extract 7-11 day variations of the semidiurnal lunar tide. The authors used the technique of Paulino et al. (2017) to derive the amplitude of the semidiurnal lunar tide. This technique involves a 27 day window, which enables to distinguish between the semidiurnal lunar tide (12.42h) and semidiurnal solar tide (12.00h). The technique should largely eliminate variations with periods less than 27 days, even though the amplitude is calculated for each day. Thus, it is unclear whether the presented short- period variations are meaningful."**

AUTHORS: We thank the reviewer for this important comment. The TEC in the tropical region is mainly produced by the absorption of the EUV and X-rays solar radiations. Thus the diurnal cycle is faraway dominant and should be removed. The determination of the lunar semidiurnal tides in TEC maps were done according to the Pedatella and Forbes (2010) methodology and only quiet days were considered ($K_p < 3$) in the analysis. Figure 1 (upper panel) shows a 29.5 days window of TEC from 27 July 2011 to 25 August 2011 measured at (15ºS, 39ºW).

After the elimination of geomagnetic influences, a Fourier analysis was performed

to extract the subharmonics of the solar day (diurnal, semidiurnal and terdiurnal oscillations). Effects of the solar rotation was removed using a 27-day window moving it forward one day at time to calculate the mean solar day centered in the window as can be seen in Figure 1 (middle panel). In addition, residual TEC was determined subtract the original TEC from the recovered one. Figure 1 (bottom panel) shows the residual TEC for this example, where the power of the diurnal cycle is reduced and other oscillations can be observed.

[Figure]

Figure 1: (Top panel) Original TEC from 27 July 2011 calculated at (15$^o$S, 39$^o$W). (Middle panel) recovered signal using sub-hamornics of the solar day within a 27 days window. (Bottom panel) Residual TEC.

In the relative residual data (residual TEC divided by the mean TEC), a least square analysis in a window of 29-day was applied using the following equation:

$$y(\tau) = \sum_{n=0}^{2} A_n \cos\left(n\tau + \phi_n\right) \tag{1}$$

where $\tau$ if the lunar time given by $\tau = t - \nu$, $\nu$ in the age of the Moon, which is set to be 0 at the New Moon. The solar time is represented by $t$, the amplitudes and phases of the lunar tide components are represented by $A_n$ and $\phi_n$, respectively.

We have included this explanation in the manuscript in order to clarify the methodology. It was requested by Reviewer #1 as well. Furthermore, the scope of the present work was investigate the day-to-day variability of the amplitude of the lunar semidiurnal tide, which was calculated for each day and change as well as shown in Pedatella and Forbes (2010) and Paulino et al. (2017). Additionally, the determination of the short-period oscillations were statistically significant as in the LS periodogram as in the wavelet analysis (the significance level were included in all plots).

REVIEWER:**" The authors are advised to check the spectrum of the original TEC data (instead of the spectrum of the semidiurnal lunar tide) to confirm that a spectral peak exists at the semidiurnal lunar tide (12.42h) as well as the sideband frequency corresponding to the quasi-10-day wave modulation of the semidiurnal lunar tide."**

AUTHORS: We thank to the reviewer for this comments. Regarding to the presence of the lunar tide in the TEC, Figure 1 of Paulino et al. (2017) shows very clear evidences in different periods.

On the other hand, we have followed the suggestion from Reviewer #2 to show the presence of the Q8D wave in the TEC. Figure 2(a) shows the data of the quiet day TEC maps for November 2013 (when the Q8D wave was strong in the amplitude of the semidiurnal lunar tide) at $8^oS$, $35^oW$ (where there are confident GNSS receivers and the amplitude of the amplitude of the Q8D was strong). One can see that there is a strong day-to-day variability in the TEC. Figure 1(b) shows the Lomb-Scargle periodogram calculated using the data from Figure 1 (a). The diurnal cycle is very pronounced compared to the other oscillation. Figure 1(c) shows the same results of Figure 1(b) but ranged from 0 to 50 PSD units. One can see that the Q8D wave peak is above of the confidence level and it demonstrates what was suggested by the reviewer.

[Figure]

Figure 2: (a) TEC data calculated to November 2013 at (8ºS and 35ºW) (b). Lomb-Scargle periodgram for the data TEC data shown in panel (a). (c) Same as Figure (b), but for zoomed to the y-range from 0 to 50 PSD.

REVIEWER:"1. Equation (1) - This needs more explanations. What is "filter" on the left-hand side? How is it applied to the data?"

AUTHORS: Thank you for the comment. We have included a citation about the application of the filter as suggested by the Reviewer #1. To apply the filter, first we apply the FFT transform, then we multiply the "filter" by the FFT signal. Finally we apply the inverse FFT to recover the filtered signal.

REVIEWER:"Lomb-Scargle periodogram - Since the authors show wavelet spectra in Figures 3 and 5-8, Lomb-Scargle periodograms in Figures 2 and 4 do not seem necessary. I suggest to remove them."

AUTHORS: The reviewer is right! Most of the aspects showed in Figure 2 and 4 can be seen in the wavelet charts. However, LS periodograms can give us a general idea about the periodicities using the whole period of analysis and comparisons between the latitudes are easily matched. Even so, if the reviewer thinks better to remove them, we can do it for the revised version.

REVIEWER:**"Figure 9 - The antisymmetric mode such as the quasi-10-day wave has the phase structure that is antisymmetric about the equator, but not the amplitude structure. That is, when there is a strong quasi-10-day wave, we should expect the amplitude of the wave to be large in both northern and southern hemispheres but with the opposite phase. What is shown in Figure 9 is the anti-correlation of the amplitude between the northern and southern hemispheres, which does not necessarily support the involvement of the quasi-10-day wave."**

AUTHORS: The reviewer is right! Figure 9 does not necessarily support the antisymmetry. Thank you for the comment. Besides, we are not sure about how is the symmetry of planetary waves regarding the magnetic coordinates. Therefore, we decide to remove these analysis. Thank you for this contribution.

[revised manuscript text omitted]

physical Research: Atmospheres, 124, 9874–9892, https://doi.org/10.1029/2019JD030634, 2019.

---

## Author Response (AR2)

[revised manuscript text omitted]

**Responses to Editor and Reviewers**

**General Comments:**

Dear Dr. Ana G. Elias.

Again, we appreciate for consider our manuscript suitable with the scope of Annales Geophysicae. We also thank the two reviewers for the comments and suggestions. Out point-by-point responses and the tracked changes manuscript can be found as following.

**Reviewer #1:**

REVIEWER:"**The authors addressed all of my concerns. I agree with reviewer #2 that the authors should "check the spectrum of the original TEC data (instead of the spectrum of the semidiurnal lunar tide) to confirm that a spectral peak exists at the semidiurnal lunar tide (12.42h) as well as the sideband frequency corresponding to the quasi-10-day wave modulation of the semidiurnal lunar tide. This analysis would add significantly to the quality of the manuscript.**"

AUTHORS: Thank your for this second round of revision. We have including this analysis in Figure 5 of this document.

**Reviewer #2:**

REVIEWER:"**Unfortunately, the authors did not address the major concerns that I raised in the previous round. My main question was how the authors can extract 7-11 day modulation of the lunar tide while the method involves a long-time window (29 days). The authors respond to this by pointing out that the technique was previously used by Pedatella and Forbes (2010). However, the study by Pedatella and Forbes (2010) mainly focused on the seasonal and longitudinal variability of the lunar tide in TEC and they did not examine variations with periods shorter than 29 days. Based on the work by Pedatella and Forbes (2010) and Paulino et al. (2017), I agree that the technique is suitable for evaluating the lunar tide in TEC and its seasonal variability. However, this does not mean that the technique can properly resolve day-to-day variability of the lunar tide. The authors also pointed out that short-period variations around 7-11 days are statistically significant in spectral analysis. This means that the quantity derived from the preprocessing includes variations around those periods, but my question is how much of these variations are actually related to the modulation of the lunar tide. The authors still need to establish that the technique can be used to extract short-period variations of the lunar semidiurnal tide.**"

AUTHORS: Again, we thank the Reviewer # 2 for spending time revising our manuscript and suggesting important points to be checked and debated. As we respond on the first round, we have used a 29-day to calculate the amplitudes of the Lunar semidiurnal tide in the residual (removing solar components and variations) TEC for the central day of the window. The reviewer is right, it is not possible to separate the lunar and solar frequencies using only a single day of measurements. However, we have shifted the 29-day

window every day to compose the daily estimations of the amplitudes of the lunar tide. We have observed that the amplitudes varying even for short period range.

The reviewer is right, Pedatella and Forbes (2010) did not do any consideration about short period variation of the lunar tide amplitudes. However, one can observe in their figures that there is short period variation (not explored by them) as well in figures of Paulino et al. (2017). The investigation of the variation in the amplitudes of the lunar semidiurnal tide is exactly the novelty of this manuscript.

REVIEWER:**"To assist the authors' response in the next round, I suggest that the authors should make a sensitivity test using synthetic data. That is, the authors could generate artificial data (time series) that contains signals associated with (1) the lunar tidal, 12.42h, variation and its 8-day modulation lasting a duration of 20 days or so, (2) the diurnal, 24h, and semidiurnal, 12h, variations and their 27-day modulations, and (3) random noise. Then, the authors could analyze the synthetic data as they did the actual data in the paper to see if the technique can resolve the 8-day modulation of 12.42h variation. I would also suggest to do a test with no 8-day modulation in the lunar semidiurnal (12.42h) component but in the solar semidiurnal (12.0h) component to make sure that the technique can distinguish the 8-day modulation of the solar and lunar semidiurnal variations."**

AUTHORS: That is a really good suggestion, we thank to the reviewer. We have performed these tests and the results are as following:

Teste 1: Artificial signal as:

$$y(t) = A_{DST} * \cos\left(\frac{2\pi}{T_{DST}}t + \phi_{DST}\right) + A_{SST} * \cos\left(\frac{2\pi}{T_{SST}}t + \phi_{SST}\right) +$$

$$A_{TST} * \cos\left(\frac{2\pi}{T_{TST}}t + \phi_{TST}\right) + A_{SLT} * \cos\left(\frac{2\pi}{T_{SLT}}t + \phi_{SLT}\right) + \tag{1}$$

$$A_{Q8DW} * \cos\left(\frac{2\pi}{T_{Q8DW}}t + \phi_{Q8DW}\right)$$

where the indexes $DST$, $SST$, $TST$, $SLT$ and $Q8DW$ represent diurnal, semidiurnal and terdiurnal solar tides, semidiurnal lunar tide and quasi 8 days wave, respectively. The amplitudes are represented by $A$, the phases by $\phi$ and the periods by $T$. We have used values for the amplitudes that are typically retrieved from the TEC data for these components.

Figure 1 summarises the Test 1. We have calculated the amplitudes for the SST and SLT using the methodology proposed in this manuscript. One can observed that, although there is a clear modulation of the TEC by the Q8D oscillation, it is not observed in the amplitudes of the tides.

**Conclusion from Test 1**: If we have the presence of the Q8DW in the TEC, it will not necessary be necessary in the amplitude of the lunar tide. Even, it is being an obvious conclusion, but it is important to the following steps of the investigations.

[Figure]

Figure 1: (a) Artificial signal generated using Equation 1. (b) Amplitudes of the semidiurnal solar tide calculated from the panel (a). (c) Periodogram for the amplitudes of the SST. (d) Amplitudes of the semidiurnal lunar tide calculated from the panel (a). (e) Periodogram for the amplitudes of the SLT. Horizontal dashed line represent the confidence level of 99%.

Teste 2: Artificial signal as:

$$y(t) = A_{DST} * \cos\left(\frac{2\pi}{T_{DST}}t + \phi_{DST}\right) +$$

$$A_{TST} * \cos\left(\frac{2\pi}{T_{TST}}t + \phi_{TST}\right) + A_{SLT} * \cos\left(\frac{2\pi}{T_{SLT}}t + \phi_{SLT}\right) + \tag{2}$$

$$A_{SST-Q8DW} * \cos\left(\frac{2\pi}{T_{SST}}t + \phi_{SST}\right)$$

where $A_{SST-Q8DW}$ is the amplitude of the semidiurnal solar tide modulated by the Q8DW.

Figure 2 (a) shows the artificial signal with a clear modulation of the Q8DW. Figure 2 (b) shows the amplitude of the SST where there is a strong Q8D variation and Figure 2 (c) shows the presence of the Q8D oscillation from the periodogram. Otherwise, Panels (d) and (f) do not show evidence of the Q8D oscillation in the amplitude of the SLT.

**Conclusion from Test 2**: If the signal has the SST modulated by the Q8DW, it will be recovered in the amplitudes of the SST and there is no leakage to the SLT.

[Figure]

Figure 2: (a) Artificial signal generated using Equation 2. (b) Amplitudes of the semidiurnal solar tide calculated from the panel (a). (c) Periodogram for the amplitudes of the SST. (d) Amplitudes of the semidiurnal lunar tide calculated from the panel (a). (e) Periodogram for the amplitudes of the SLT. Horizontal dashed line represent the confidence level of 99%.

Teste 3: Artificial signal as:

$$y(t) = A_{DST} * \cos\left(\frac{2\pi}{T_{DST}}t + \phi_{DST}\right) + A_{SST} * \cos\left(\frac{2\pi}{T_{SST}}t + \phi_{SST}\right) +$$

$$A_{TST} * \cos\left(\frac{2\pi}{T_{TST}}t + \phi_{TST}\right) + A_{SLT-Q8DW} * \cos\left(\frac{2\pi}{T_{SLT}}t + \phi_{SLT}\right)$$

(3)

where $A_{SLT-Q8DW}$ is the amplitude of the semidiurnal lunar tide modulated by the Q8DW.

Figure 3 (a) shows also a roughly modulation of the Q8DW in the TEC. Figure 3 (b) and (c) do not show any significant Q8DW periodicity. Meanwhile, Panels (d) and (f) show the presence of the Q8D oscillation in the amplitude of the SLT.

**Conclusion from Test 3**: If the signal has the SLT modulated by the Q8DW, it will be recovered in the amplitudes of the SLT and there is no leakage to the SST.

[Figure]

Figure 3: (a) Artificial signal generated using Equation 3. (b) Amplitudes of the semidiurnal solar tide calculated from the panel (a). (c) Periodogram for the amplitudes of the SST. (d) Amplitudes of the semidiurnal lunar tide calculated from the panel (a). (e) Periodogram for the amplitudes of the SLT. Horizontal dashed line represent the confidence level of 99%.

From the tests above we can confirm that our methodology is enough to isolate the SLT from the SST. We decided to go further and test the real data. We have chosen an interval when the Q8D oscillation appeared strong in the amplitude of the SLT and chosen the most southern pointer of the the Brasil where the SST is generally strong as compared to the equatorial latitude and applied the same methodology, the result are presented in Test 4.

Teste 4: Real observed TEC from February to April 2014 at $30^oS$ and $54^oW$.

Figure 4 shows the test made using measured TEC. Panel (a) shows an aparente variation of Q8D in the whole data. Panels (b) and (c) do not show significative amplitudes around Q8D in the amplitudes of the SST. However, in Panels (d) and (e), the amplitudes of the SLT showed strong Q8D oscillation. it corroborates with the results shown in Figure 8 of the manuscript. One can observed that there is no leakage associated with SST.

**Conclusion from Test 4**: Indeed the present short variation around Q8D in the amplitude of the SLT comes from the interaction of this oscillation with theses waves and there is no leakeage associated with the solar component.

Based on the relevance of this conclusion, we have added this Figure 4 to the manuscript and re-written some statements. Again, we thank to the reviewer for this important contribution to our manuscript.

[Figure]

Figure 4: (a) Real TEC collected at 30ºS and 54ºW from February to April 2014. (b) Amplitudes of the semidiurnal solar tide calculated from the panel (a). (c) Periodogram for the amplitudes of the SST. (d) Amplitudes of the semidiurnal lunar tide calculated from the panel (a). (e) Periodogram for the amplitudes of the SLT. Horizontal dashed line represent the confidence level of 99%.

REVIEWER:"In my previous comments, I also suggested that the authors should check the spectrum of the original TEC data and confirm the presence of spectral peaks corresponding to Q8DW modulation of the lunar tide. The authors, instead of showing spectral peaks for the Q8DW modulation of the lunar tide, presented a peak corresponding to the Q8DW itself. "

AUTHORS: Figure 2 (c) of the former responses shows the periodogram for the original TEC and not for the SLT amplitude. There is a clear the presence of the Q8D oscillation above the 99% confidence level.

REVIEWER:"Since it appears that the authors did not understand what I meant, here I clarify my points. The lunar tide has a frequency of f1 = 1.93 (= 1/0.5175) cycle/day. On the other hand, the Q8DW has a frequency of f2 = 0.125 (=1/8) cycle/day. The Q8DW modulation of the lunar tide should generate spectral components at frequencies of the sum and difference of these frequencies, that is, f1+f2 = 2.055 cycle/day (or 11.68h in period) and f1-f2 = 1.805 cycle/day (or 13.30h). If the Q8DW modulation of the lunar tide is truly detected, the authors should be able to find spectral peaks at periods of 11.68h and/or 13.30h. However, in Figure 2 of the authors' response letter, there is no significant peak at periods around 11-13h. This means that the technique used was not able to resolve the Q8DW modulation of the lunar tide, at least for November 2013. I would also like to point out that, if there is Q8DW modulation of the solar semidiurnal, 12h, variation, the expected spectral peaks are at 11.29h and 12.80h."

AUTHORS: Although complex, this is an interesting point to investigate on the non-linear interaction between tides and PWs. First of all, we agree with the reviewer that it is difficult to see any thing around 0.45 and 0.55 day in Figure 2 of the previous responses, but it was out the scope of the previous comment where the reviewer would like to see Q8D oscillation in the TEC. Additionally, we have decided to zoom around the 2 cycles per day to reply this question. Figure 5 (a) shows the real TEC collected at 25$^o$S and 49$^o$W from February to April 2014. These data showed in Panel (b) two peaks around 7 and 9 days above the confidence level and the SLT (Panel c) is above as well. We have used the theory mentioned by the reviewer and marked the predicted position for the secondary peaks in Panel (c) for both oscillations (Q7D and Q9D) supposedly interacting non-linearly with SLT.

One can observed that even below the confidence level, there are peaks almost compatible with the non-linear interaction of the Q7D and Q9D oscillation with the SLT. Only two symmetrical peaks were observed around the SST, which can be associated to a likely interaction with Q4D oscillation. In addition, there is no corespondent peaks to the non-linear interaction of the Q7D and Q9D with SST. We understand that more investigation is necessary to reach relevant conclusion on this point, aspect like where effectively the non-linear interaction occur in the atmosphere, how long time of interaction is necessary to observed peaks around the confidence levels of the periodgrams, and other points must be further studied. However, it is out of the scope of the present manuscript.

[Figure]

Figure 5: (a) Real TEC collected at 35ºS and 49ºSW from February to April 2014. (b) Periodogram showing periods from 2 to 12 days. (c) Periodogram zoomed around the 2 cycles per day oscillation.

REVIEWER:"**Thus, it would be challenging to distinguish between the Q8DW modulations of the solar and lunar semidiurnal variations. I suspect that what the authors observed could be the result of the Q8DW modulation of the solar semidiurnal tide, rather than the Q8DW modulation of the lunar semidiurnal tide.**"

AUTHORS: We hope to be addressed this concerns of the Reviewer #1 after this revision.